# Evaluating LLMs When They Do Not Know the Answer: Statistical Evaluation of Mathematical Reasoning via Comparative Signals

Zihan Dong [* 1]  Zhixian Zhang [* 1]  Yang Zhou [1]  Can Jin [1]  Ruijia Wu [2]  Linjun Zhang [1]

## Abstract

Evaluating mathematical reasoning in LLMs is constrained by limited benchmark sizes and inherent model stochasticity, yielding high-variance accuracy estimates and unstable rankings across platforms. On difficult problems, an LLM may fail to produce a correct final answer, yet still provide reliable pairwise comparison signals indicating which of two candidate solutions is better. We leverage this observation to design a statistically efficient evaluation framework that combines standard labeled outcomes with pairwise comparison signals obtained by having models judge auxiliary reasoning chains. Treating these comparison signals as control variates, we develop a semiparametric estimator based on the efficient influence function (EIF) for the setting where auxiliary reasoning chains are observed. This yields a one-step estimator that achieves the semiparametric efficiency bound, guarantees strict variance reduction over naive sample averaging, and admits asymptotic normality for principled uncertainty quantification. Across simulations, our one-step estimator substantially improves ranking accuracy, with gains increasing as model output noise grows. Experiments on GPQA Diamond, AIME 2025, and GSM8K further demonstrate more precise performance estimation and more reliable model rankings, especially in small-sample regimes where conventional evaluation is pretty unstable.

## 1. Introduction

Mathematical reasoning has emerged as a critical benchmark for assessing the genuine cognition of LLMs (Yan et al., 2025; Ahn et al., 2024). This ability is typically quan-

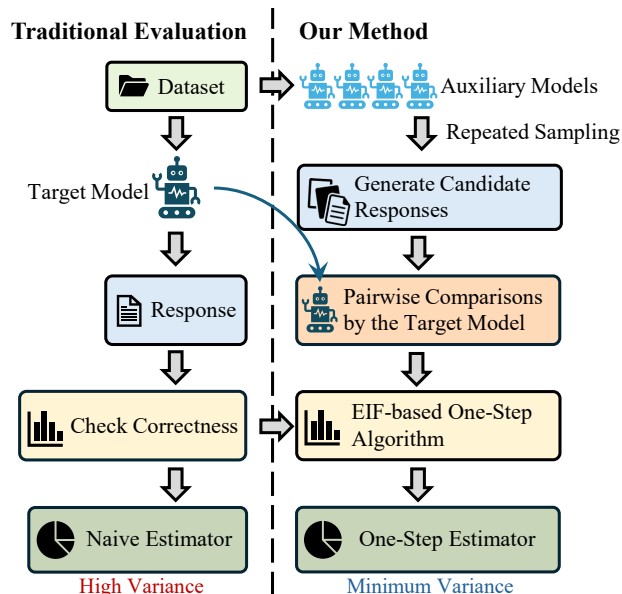

*Figure 1.* Overview of our Semiparametric Evaluation Framework. We augment standard accuracy evaluation with pairwise comparison signals and construct an EIF-based one-step estimator that achieves the semiparametric efficiency bound, yielding strict variance reduction.

tified through accuracy-based performance, irrespective of the specific metrics employed. However, despite its importance for high-stakes deployment, current evaluations suffer from a reproducibility crisis, characterized by substantial discrepancies in reported accuracy metrics across different evaluation platforms—even within the domain of mathematical reasoning, where objective ground truths should guarantee consistency. For instance, regarding the DeepSeek-R1-Distill-Llama-70B model, Guo et al. (2025) reports a GPQA Diamond pass@1 accuracy of 65.2%, whereas independent evaluation on Artificial Analysis reports 40.2%, and for Epoch AI reports 55.7%, and our own replication yields 59.09%.

This variation stems from two primary factors. First, LLMs exhibit high intrinsic stochasticity, especially on challenging math problems, leading to instability and significant output variance (Sun et al., 2025). Second, mathematical benchmarks are severely size-constrained. For example, AIME 2025 contains only 30 problems, implying that a single cor-

*Equal contribution [1]Rutgers University [2]Shanghai Jiao Tong University. Correspondence to: Linjun Zhang <linjun.zhang@rutgers.edu>.

*Proceedings of the 43rd International Conference on Machine Learning*, Seoul, South Korea. PMLR 306, 2026. Copyright 2026 by the author(s).

rect or incorrect response shifts the reported accuracy by more than three percentage points. In such small-sample regimes, stochastic variation dominates signal, rendering naive accuracy estimates noisy and model rankings brittle.

To formalize this challenge, we frame the accuracy measurement as an unknown population parameter to be estimated from noisy and limited observations. From a statistical perspective, estimator precision is fundamentally governed by the amount of information extracted from the available data. Because expanding benchmark sizes is often infeasible, particularly for expert-curated mathematical datasets, improving evaluation reliability requires extracting more information per sample, rather than collecting additional samples. This perspective motivates a central question: *Where can we obtain additional information beyond the target model's final answers, and how can this information be leveraged to improve the statistical efficiency of accuracy estimation?*

A key empirical observation motivates our approach. On difficult mathematical problems, an LLM may fail to produce a correct final answer, yet still exhibit a reliable ability to compare two candidate solutions and identify which one is better, known as the *generation-verification gap* (Christiano et al., 2017; Stiennon et al., 2020; Ouyang et al., 2022; Rafailov et al., 2024). In other words, while absolute correctness is noisy and sparse, relative judgments about solution quality are often substantially more stable.

We leverage this observation to design a statistically efficient evaluation framework that combines standard, possibly noisy, labeled outcomes with pairwise comparison signals. Specifically, we obtain auxiliary information by having models judge paired reasoning chains generated by auxiliary models, and treat these comparison signals as additional side information to improve the efficiency of the accuracy estimation. The procedure is summarized in Fig. 1.

Our main contributions are summarized as follows:

1. **Semiparametric Evaluation Framework.** We formalize LLM accuracy evaluation as a statistical inference problem and propose a novel methodology that leverages pairwise comparison signals derived from the target model's evaluation of auxiliary responses as control variates. Using tools from semiparametric inference, we derive the Efficient Influence Function (EIF) under a semiparametric model where the auxiliary generation mechanism is known, and develop a practical One-Step Estimator with cross-fitting. For small-sample regimes, we further introduce an in-context learning approach that utilizes off-the-shelf LLMs as semantic regressors.

2. **Theoretical Guarantees.** We show that the proposed estimator is asymptotically normal and attains the semiparametric efficiency bound, enabling principled uncertainty quantification for LLM evaluation with the optimal efficiency. The efficiency gain is strictly positive whenever the auxiliary comparison signals provide non-redundant information beyond the original inputs.

3. **Empirical Validation.** We validate our framework on both simulated data and three real-world benchmarks (GPQA Diamond, AIME 2025, GSM8K), demonstrating consistent variance reduction and superior model evaluation accuracy compared to the naive sample mean estimator.

## 1.1. Related Work

**Efficiency and Rigor in LLM Evaluation.** LLM evaluation has evolved from static benchmarks to dynamic, model-based paradigms (Chang et al., 2024; Ni et al., 2025; Lu et al., 2023; Zheng et al., 2023; Liu et al., 2025b; Ibrahim et al., 2025), yet most work neglects *statistical efficiency* (Boyeau et al., 2024). Recent efforts address estimation variance via control variates (Frick et al., 2025; Zhou et al., 2025), Bayesian refinement (Hariri et al., 2025), and pairwise graphs with uncertainty quantification (Mahdavi et al., 2025; Chernyshev et al., 2025). Typically, in small-sample settings (e.g., AIME with 30 problems), Bayesian approaches can be highly sensitive to prior specification, and their performance may vary substantially without a well-justified prior. Moreover, these lack unified theoretical grounding. We address this gap by treating evaluation as information extraction: mining auxiliary reasoning signals to achieve semiparametric efficiency.

**Pairwise Comparisons and Alignment Signals.** Pairwise comparisons underpin modern alignment methods including RLHF and DPO (Christiano et al., 2017; Stiennon et al., 2020; Ouyang et al., 2022; Rafailov et al., 2024), valued because verification is simpler than generation (Cobbe et al., 2021; Lightman et al., 2023) and such data is cheaply elicited via LLM judges (Zheng et al., 2023; Dong et al., 2026). Beyond training, pairwise signals enable inference-time consistency checks (Khan et al., 2024; Chen et al., 2023) and robust preference aggregation (Mahdavi et al., 2025; Chernyshev et al., 2025). We argue that this data constitutes an underutilized auxiliary source for evaluation: the discriminative signal from verifying reasoning chains is highly correlated with ground-truth correctness, making it an ideal control variate.

**Semi-parametric Inference.** Semiparametric inference provides a general framework for achieving optimal estimation efficiency by leveraging auxiliary information as control variates (Robins et al., 1994; Robins & Rotnitzky, 1995; Tsiatis, 2006; van der Vaart, 1998). Central to this framework is the Efficient Influence Function (EIF) (Kennedy, 2022), which characterizes the optimal variance achievable by any regular estimator and guides the construction of one-step corrected estimators.

## 2. Problem Set-up

Consider evaluating a target LLM on mathematical reasoning tasks. Let $\mathcal{T}$ denote the space of texts. For each problem instance, let $X \in \mathcal{T}$ denote the input prompt, $Y \in \mathcal{T}$ the model's final response, and $G \in \mathcal{T}$ the ground-truth answer. We define the model performance through a metric function $\phi : \mathcal{T} \times \mathcal{T} \to \mathbb{R}^+$. Model performance is quantified through a metric function:

$$\theta := \mathbb{E}[\phi(Y, G)].$$

For accuracy-based evaluation, particularly in mathematical reasoning tasks where exact correctness is required, the metric function takes the form $\phi_{\mathrm{acc}}(y, g) = \mathbb{1}\{y = g\}$, where $\mathbb{1}\{\cdot\}$ is the indicator function. Thus, $\theta$ represents the probability that the model's response exactly matches the ground truth. We use $(x, y, g)$ to denote the observed realizations of $(X, Y, G)$.

Formally, evaluating the LLM's capability can be cast as a statistical inference problem for the target parameter $\theta$. We assume access to a primary evaluation dataset of $N$ i.i.d. observations $\mathcal{D} = \{(x_i, y_i, g_i)\}_{i=1}^N$. The goal is to estimate $\theta$. The standard approach relies solely on the ground-truth $G$ and the answer $Y$, yielding the *naive estimator*:

$$\widehat{\theta}^{\mathrm{naive}} = \frac{1}{N} \sum_{i=1}^N \phi(y_i, g_i). \tag{2.1}$$

While unbiased, this estimator can exhibit high variance in small-sample regimes, which are common in mathematical benchmarks.

To achieve more efficient estimation, specifically to construct an estimator with smaller variance than $\widehat{\theta}^{\mathrm{naive}}$, we propose to leverage auxiliary information. Our key insight is based on the observation that verifying a solution is often less demanding than generating one (Cobbe et al., 2021; Lightman et al., 2023), and therefore the discriminative signal from a judge provides valuable auxiliary information. Even when a model fails to produce a correct final answer, its relative judgments over candidate solutions may still be informative about its reasoning competence. The use of pairwise comparisons also aligns with the increasing adoption of pairwise comparison data in recent LLMs literature (Bradley & Terry, 1952; Zheng et al., 2023; Dubois et al., 2024; Rafailov et al., 2024). Formally, beyond the primary evaluation data $\mathcal{D}$, we assume access to auxiliary signals for each problem instance $i \in [N]$ in the form of

$$z_{ij} = (w_{1ij}, w_{2ij}, v_{ij}),$$

where index $j$ denotes the $j$-th auxiliary generation. Here, $w_{1ij}, w_{2ij} \in \mathcal{T}$ are two candidate reasoning chains (including final responses) generated by auxiliary LLMs, which may coincide with or differ from the target model, and $v_{ij}$ is

a pairwise comparative signal produced by the target model indicating which candidate is preferred. In the next section, we show how these auxiliary signals $Z = (W_1, W_2, V)$ can be systematically incorporated to improve the efficiency of estimating $\theta$.

## 3. Method

Our primary objective is to improve upon the statistical efficiency of the naive baseline $\widehat{\theta}^{\mathrm{naive}}$ by incorporating the rich auxiliary signals $Z$. We formalize this by treating these signals as *control variates*, that is, variables whose distribution is fully known and can be repeatedly sampled, within the semiparametric framework (Deng et al., 2013; Tsiatis, 2006; Kennedy, 2022; Hines et al., 2022).

Intuitively, if the auxiliary signals are correlated with the ground truth, we can query the models to extract this additional information, thereby reducing the variance of our inference on $\theta$. Different from the traditional semiparametric theory for observational studies, in LLM evaluation, the auxiliary signals $Z$ are generated purely by the LLMs (including target and auxiliary models). Consequently, their conditional distribution given the input $X$ is specified by the generation mechanism and can be approximated arbitrarily well via repeated Monte Carlo sampling. We leverage this property to construct an estimator with smaller variance.

### 3.1. Efficient Influence Function

To construct an estimator with optimal statistical properties, we derive the Efficient Influence Function (EIF) for our target parameter $\theta$. We consider the observations $(X, Y, G, Z)$, where $Z = (W_1, W_2, V)$ denotes the auxiliary signals.

*Definition* 3.1 (Semiparametric Model Space for AI Evaluation). We define the semiparametric model space for AI evaluation as

$$\mathcal{M} = \left\{ \begin{array}{l} P : p(y, g, z, x) = p(y, g \mid z, x)\, p(z \mid x)\, p(x), \\ \quad p(z \mid x) \text{ is known and fixed by the evaluation protocol} \end{array} \right\}.$$

Equivalently, the model is nonparametric in $p(y, g \mid z, x)$ and $p(x)$, while $p(z \mid x)$ is treated as known and therefore does not vary over $\mathcal{M}$.

*Remark* 3.2. In our AI evaluation setting, the condition that $p(z \mid x)$ is known is not an additional statistical assumption; it is part of the model specification induced by the experimental design. The auxiliary labels are generated on-demand via auxiliary and target models under a fixed prompting mechanism, so this conditional distribution can be accessed, up to arbitrary precision, through repeated Monte Carlo sampling.

Based on Definition 3.1, we derive the EIF for the target parameter $\theta$ under this AI evaluation model space.

*Proposition* 3.3 (Efficient Influence Function with Known

Conditional Distribution). Under the semiparametric model space $\mathcal{M}$ in Definition 3.1, the EIF for $\theta$ evaluated at a data point $(X, Y, G, Z)$ is given by:

$$\psi(X, Y, G, Z) = \big((m(X) - \theta)\big) - \big(\tau(X, Z) - \phi(Y, G)\big) \tag{3.1}$$

where the nuisance functions $\tau$ and $m$ [1] satisfy

$$\tau(X, Z) = \mathbb{E}[\phi(Y, G) \mid Z, X], \tag{3.2}$$

$$m(X) = \mathbb{E}[\phi(Y, G) \mid X] = \int \tau(X, z)\, dP(z \mid X). \tag{3.3}$$

*Remark* 3.4. The structure of the EIF directly reflects the orthogonal decomposition of the tangent space (see Appendix A):

- **Outcome regression** ($\tau$): The function $\tau(x, z)$ represents the expected correctness of the target LLM given the problem $x$ and observed auxiliary signals $z = (w_1, w_2, v)$. It learns to predict whether the target model answers correctly by leveraging auxiliary data $z$. This enables evaluation even when ground-truth labels $Y$ are unavailable, as the auxiliary signals provide informative proxies for the target model's performance.

- **Integrated regression** ($m$): The function $m(x)$ marginalizes over all possible auxiliary LLM generations. A key advantage of LLM evaluation is that we can query auxiliary models arbitrarily many times on the same problem $x$, allowing us to approximate $m(x)$ arbitrarily well via repeated Monte Carlo sampling as shown in Equation (3.4).

By construction, $m(X)$ is the $Z \mid X$-marginalization of $\tau(X, Z)$; hence $m(X) = \mathbb{E}[\tau(X, Z) \mid X]$ and $\mathbb{E}[m(X) - \tau(X, Z)] = 0$. We also note that the two terms $(\phi(Y, G) - \tau(X, Z))$ and $(m(X) - \theta)$ in Equation (3.1) are orthogonal (the detailed derivation is deferred to Appendix A.4). This orthogonality, combined with informative auxiliary signals, yields a strictly lower asymptotic variance than the naive estimator.

### 3.2. One-Step Estimator

To translate the theoretical EIF into a practical algorithm, we propose an *One-Step Estimator* Algorithm. Specifically, we utilize Cross-Fitting (Chernozhukov et al., 2018) where the nuisance function $\widehat{\tau}$ evaluated at a data point $X_i$ is trained on a partition independent of $i$, thereby decoupling the nuisance estimation from the inference of $\theta$. We then calibrate the naive estimator by fitting two nuisance components based on the form of EIF in (3.1).

The complete procedure is detailed in Algorithm 1. We approximate the integration in (3.3) via Monte Carlo sampling,

---

[1] Nuisance function is a commonly used term in semiparametric inference. For self-completeness, we introduce its formal definition in Appendix D.

utilizing the known generator $p(z \mid x)$.

---

**Algorithm 1** EIF based One-Step Algorithm

---

1: **Input:** Dataset $\mathcal{D}$ with $M + 1$ auxiliary samples per instance:

$$\mathcal{D} = \Big\{ x_i, y_i, g_i, \{(w_{1ij}, w_{2ij}, v_{ij})\}_{j=1}^{M+1} \Big\}_{i=1}^{N}.$$

2: Randomly partition indices $\{1, \ldots, N\}$ into $K$ disjoint folds $I_1, \ldots, I_K$.
3: **for** $k = 1, \ldots, K$ **do**
4:   Let $I_k^c$ denote the training set (all indices except $I_k$).
5:   **(Outcome regression)** Use data in $I_k^c$ to fit $\widehat{\tau}^{(-k)}(x, z)$ as an estimate of $\tau(x, z) = \mathbb{E}[\phi(Y, G) \mid X = x, Z = z]$, where $z = (w_1, w_2, v)$.
6:   **(Integrated regression)** For any $x$, compute using the last $M$ samples:

$$\widehat{m}^{(-k)}(x) = \frac{1}{M} \sum_{j=2}^{M+1} \widehat{\tau}^{(-k)}(w_{1j}, w_{2j}, v_j, x). \tag{3.4}$$

7:   **for** $i \in I_k$ **do**
8:     Compute influence score using the first auxiliary sample ($j = 1$):

$$\widehat{\psi}_i^{(-k)} = \widehat{m}^{(-k)}(X_i) + \phi(Y_i, G_i) - \widehat{\tau}^{(-k)}(w_{1i,1}, w_{2i,1}, v_{i,1}, X_i).$$

9:   **end for**
10: **end for**
11: **Output:**

$$\widehat{\theta}^{\text{1-step}} = \frac{1}{N} \sum_{k=1}^{K} \sum_{i \in I_k} \widehat{\psi}_i^{(-k)}. \tag{3.5}$$

---

*Remark* 3.5 (Semantic Regression for Small Samples). In small-sample regimes (e.g., AIME25 with $N = 30$), training a high-dimensional regression model $\widehat{\tau}(x, z)$ is infeasible. Instead, we approximate $\tau(x, z)$ by leveraging the reasoning capabilities of off-the-shelf LLMs (e.g., Gemini-3-Flash) via in-context learning. We treat the LLM as a fixed "semantic regressor" that predicts the probability of correctness based on the problem context. This approach offers two key advantages: (1) **no training cost**: since the LLM's predictive capability derives entirely from pre-trained weights rather than gradient updates on the evaluation set, we avoid the computational burden of model fitting altogether. (2) **statistical independence**: because the semantic regressor is fixed *a priori* and independent of the evaluation samples, cross-fitting becomes unnecessary, further simplifying the estimation pipeline. Empirically, this strategy performs remarkably well even with extremely small sample sizes, as the pre-trained LLM effectively transfers its broad reasoning capabilities to the nuisance estimation task. The adapted procedure is detailed in Algorithm 2 in Appendix C.3.

# 4. Theoretical Analysis

We establish two main theoretical results for our one-step estimator. First, we prove asymptotic normality and show that it achieves the semiparametric efficiency bound under mild regularity conditions (Theorem 4.5). Second, we quantify the variance reduction relative to the naive sample mean and establish conditions under which strict efficiency gains are guaranteed (Corollary 4.7).

## 4.1. Asymptotic Normality

*Assumption* 4.1 (Moment Condition). The evaluation metric $\phi(Y, G)$ has finite second moments, i.e., $\mathbb{E}[\phi(Y, G)^2] < \infty$.

*Assumption* 4.2 (Consistency). $\widehat{\tau}$ converges to $\tau$ in $L_2$ norm:

$$\|\widehat{\tau} - \tau\|_{P_{(X,Z)},2} = \sqrt{\mathbb{E}[(\widehat{\tau}(X, Z) - \tau(X, Z))^2]} = o_p(1),$$

where $P_{(X,Z)}$ denotes the marginal distribution of $(X, Z)$ under $P$, and $\|f\|_{P,2} = \sqrt{\mathbb{E}_P[f^2]}$ denotes the $L_2$ norm with respect to the probability measure $P$.

*Assumption* 4.3 (Monte Carlo Approximation). $M \to \infty$ as $N \to \infty$, ensuring the approximation error is negligible relative to $N^{-1/2}$.

*Remark* 4.4. Assumption 4.1 ensures finite variance for semiparametric inference. It is trivially satisfied for accuracy metrics ($\phi \in \{0, 1\}$) and holds broadly for any bounded evaluation metric. Assumption 4.2 guarantees that the nuisance estimator converges to the true conditional expectation. Notably, this requirement is substantially weaker than that in standard semiparametric inference (Chernozhukov et al., 2018), which demands convergence at rate $N^{-1/4}$; thus, even flexible machine learning methods (e.g., neural networks, random forests) with slow convergence rates can be employed. Assumption 4.3 controls the Monte Carlo approximation error in computing $m(X)$. It is easily satisfied since $P(Z \mid X)$ is known and we can generate arbitrarily many auxiliary samples at negligible cost.

*Theorem* 4.5 (Asymptotic Normality). Under Assumptions 4.1–4.3, $\widehat{\theta}_{\text{1-step}}$ is asymptotically normal:

$$\sqrt{N}(\widehat{\theta}_{\text{1-step}} - \theta) \xrightarrow{d} \mathcal{N}(0, \sigma^2_{\text{eff}}), \qquad (4.1)$$

where $\sigma^2_{\text{eff}} = \text{Var}(\psi(X, Y, G, Z))$. Furthermore, $\widehat{\theta}_{\text{1-step}}$ achieves the semiparametric efficiency bound in the sense that no other regular estimators have a smaller variance than $\widehat{\theta}_{\text{1-step}}$.

*Remark* 4.6. Theorem 4.5 establishes that our estimator is $\sqrt{N}$-consistent and asymptotically normal. This statistical rigor is vital for LLM evaluation, where small benchmarks (e.g., AIME) often yield unstable rankings; valid confidence intervals allow us to distinguish genuine improvements from stochastic noise. Furthermore, our framework is highly general: the auxiliary variable $Z$ can encompass any signal (e.g., Likert scores, multi-way rankings) beyond pairwise comparisons. Crucially, our framework facilitates valid evaluation regardless of the target model's large output variability. Even in high-noise regimes, our approach leverages auxiliary correlations to "calibrate" the estimate, ensuring reduced variance where naive estimators fail.

## 4.2. Variance Reduction

We quantify the efficiency gain compared to the naive estimator (2.1). Let $\sigma^2_{\text{naive}} = \text{Var}(\phi(Y, G))$ and $\sigma^2_{\text{eff}}$ denote asymptotic variances.

*Corollary* 4.7 (Strict Efficiency Gain). Assume that the auxiliary variable $Z$ is **not conditionally independent** of the target metric $\phi(Y, G)$ given $X$, i.e., $\tau(X, Z) \neq m(X)$ holds with positive probability. Then, the one-step estimator strictly reduces the asymptotic variance:

$$\sigma^2_{\text{eff}} < \sigma^2_{\text{naive}}.$$

*Remark* 4.8. The condition $\tau(X, Z) \neq m(X)$ assumes that auxiliary labels contain "novel" information, that is, the generation-verification gap is nonzero. If $Z$ were independent of the outcome given $X$, knowing $Z$ does not contribute to the estimation and our estimator reduces to the naive estimator. In practice, since $Z$ is also partially obtained by the target model, this independence is naturally violated, ensuring efficiency gain.

# 5. Experiments

This section presents our experimental evaluation. In Section 5.1, we conduct a controlled simulation study where the ground truth of model capabilities is known, enabling us to validate our theoretical findings and systematically investigate the relationship between auxiliary information quality and estimation efficiency. In Section 5.2, we apply our framework to real-world mathematical reasoning benchmarks, including GPQA (Rein et al., 2024), AIME 2025, and GSM8K (Cobbe et al., 2021), to demonstrate its practical effectiveness.[2]

## 5.1. Simulation Study

We conduct a controlled simulation study to validate our theoretical results and evaluate the practical performance of our proposed estimator. Specifically, we consider $L = 3$ target models, $N = 1000$ evaluation samples, and $M = 500$ Monte Carlo samples per instance. For each sample $i \in [N]$ and model $l \in [L]$, we observe the input $X_i$, model output $Y_{li}$, ground truth $G_i$, and auxiliary information $\{(W_{1,lij}, W_{2,lij}, V_{lij})\}_{j=1}^{M+1}$ consisting of $M+1$ i.i.d. draws of two auxiliary reasoning responses and a preference label. Our goal is to estimate the expected loss $\theta_l = \mathbb{E}[\phi(Y_{li}, G_i)]$ for each model and produce an accurate ranking of model

---

[2]Code available at: https://github.com/zihandong02/AI_evaluation

performance. By controlling the underlying parameters, we systematically study how auxiliary information quality affects estimation efficiency and ranking accuracy, demonstrating that our one-step estimator yields more reliable model comparisons than the naive baseline under constrained sample sizes.

### 5.1.1. DATA GENERATION

**Data Generative Model.** For sample $i = 1, \ldots, N$, the input $X_i$ is generated from a standard Gaussian distribution $X_i \sim \mathcal{N}(0, 1)$, and the ground truth $G_i$ are set to $G_i = X_i$. The output $Y_{li}$ of model $l$ is generated linearly with a model-specific signal level $\sigma_l$: $Y_{li} = X_i + \epsilon_{li}$, where the model-specific signal $\epsilon_{li} \sim \mathcal{N}(0, \sigma_l^2)$ is independent of $X_i$, representing the intrinsic information of the model output.

We evaluate the performance using the squared error metric $\phi(y, g) = (y - g)^2$. The target parameter $\theta_l$ represents the Mean Squared Error (MSE) of model $l$: $\theta_l = \mathbb{E}[\phi(Y_{li}, G_i)]$.

**Auxiliary Information.** For each auxiliary draw $j \in [M + 1]$, we simulate two auxiliary responses $W_{1,lij}, W_{2,lij}$ and a preference label $V_{lij}$. The noise terms $\eta_{s,ij} \sim \mathcal{N}(0, \sigma_\eta^2)$ are i.i.d. for $s \in [2]$, $i \in [N]$, $j \in [M + 1]$, and the auxiliary responses are generated by $W_{s,lij} = X_i + \rho_s \epsilon_{li} + \eta_{s,ij}$ where the correlation coefficients $\rho_1, \rho_2$ control the auxiliary information quality for each response. Crucially, the auxiliary responses are dependent through the output information $\epsilon_{li}$, reflecting the correlation of the model's internal state. The preference label is generated as $V_{lij} = \mathbb{1}\{|W_{1,lij} - Y_{li}| \le |W_{2,lij} - Y_{li}|\}$.

The key design insight is that the shared latent component $\epsilon_{li}$ represents stochastic reasoning variability that influences both the final output and the auxiliary responses. This dependence structure ensures that the auxiliary signals are informative about the target loss, enabling substantial variance reduction when incorporated as control variates. As a result, the proposed estimator can achieve accurate estimation even with noisy and limited samples.

**Efficient Influence Function.** According to equation (3.1), we obtain the EIF in this setting:

$$\psi(X, Y, G, Z) = (Y - G)^2 - (1 - \kappa\rho)\sigma_l^2 - \kappa^2(W - X)^2,$$

where $\kappa = \rho\sigma_l^2 / (\rho^2\sigma_l^2 + \sigma_\eta^2)$ (see Appendix B.1 for the complete derivation).

### 5.1.2. EXPERIMENTAL SETUP

While point estimation accuracy is a standard metric, we shift our primary focus to ranking accuracy, specifically measured by Exact Match Accuracy and Kendall's Tau. This shift is motivated by the fact that in practical model evaluation, the primary goal is often reliable model selection rather than precise scoring. By leveraging the conditional

dependence between the auxiliary information $W$ and target output $Y$, our one-step estimator converts variance reduction into the concept of sample efficiency, allowing the true model hierarchy to emerge even with limited samples where naive baselines fail to distinguish between candidates.

**Experimental Procedure.** To evaluate ranking performance, we conduct $R = 100$ independent trials where datasets are generated for models $l \in \{1, 2, 3\}$. In each trial, we compute both the naive and one-step estimates, $\{\widehat{\theta}_l^{\text{naive}}, \widehat{\theta}_l^{\text{1-step}}\}$, and derive the estimated ranking $\widehat{\pi}$ by sorting these values (i.e., $\widehat{\pi} = \text{argsort}(\widehat{\theta}_1, \widehat{\theta}_2, \widehat{\theta}_3)$). Since the ground truth parameters are set such that $\theta_1 < \theta_2 < \theta_3$, the true ranking is $\pi^* = (1, 2, 3)$. We measure the alignment between $\widehat{\pi}$ and $\pi^*$ using two metrics: (1) *Exact Match Accuracy*, $\text{Acc}_{\text{exact}} = \frac{1}{R} \sum_{r=1}^{R} \mathbb{1}\{\widehat{\pi}^{(r)} = \pi^*\}$, which captures the probability of recovering the perfect ordering; and (2) *Kendall's Tau* ($\tau$), which quantifies the rank correlation based on the number of concordant ($n_c$) and discordant ($n_d$) pairs.

### 5.1.3. MAIN RESULTS AND ANALYSIS

We consider evaluating $L = 3$ models with distinct error levels, indexed by $l \in \{1, 2, 3\}$. For the auxiliary information, we set correlation coefficients $\rho_1 = 0.8$ and $\rho_2 = 0.6$ respectively, and the auxiliary noise term $\sigma_\eta = 0.6$. We set the number of folds in cross-fitting to $K = 5$.

**Model-specific Signal.** We first conduct a sensitivity analysis on the ranking accuracy relative to the model-specific signal $\sigma_l^2$. We maintain a constant variance gap of $0.05$ between the three models while incrementally shifting the base variance, testing whether the estimators can still resolve the true model hierarchy when the output noise becomes increasingly dominant. From Figure 2, we observe that an

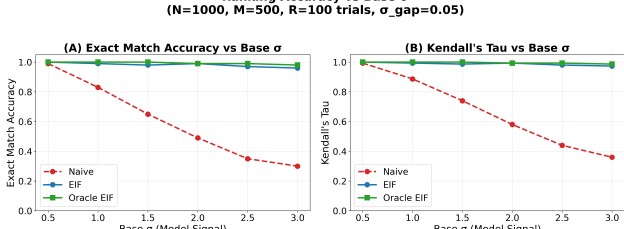

*Figure 2.* Ranking accuracy vs. model-specific signal

increase in the output variance $\sigma_l^2$ leads to a sharp decline in the ranking accuracy of the Naive estimator. In contrast, the proposed one-step estimator remains comparatively stable and preserves a higher level of ranking accuracy across a wide range of noise levels with the same number of samples. Notably, the performance of our EIF-based estimator closely tracks the "**Oracle**" curve, which utilizes the theoretical $m$, despite using only simulated approximations for $m$. This

high degree of overlap validates the precision of our EIF fitting and confirms that our estimator effectively captures the underlying model characteristics even in noisy regimes. Moreover, the performance gap between the two methods widens as noise becomes more dominant, highlighting the robustness of our approach and its sample efficiency in handling increasingly noisy and complex settings. We defer to Appendix B.3 a detailed discussion of why ranking accuracy decreases as $\sigma_l^2$ increases.

**Auxiliary Noise.** We also investigate how the information density of auxiliary variables influences estimation gains by varying the intensity of auxiliary noise. Even as the predictive power of auxiliary responses diminishes, the one-step estimator consistently extracts sufficient signal to outperform the naive baseline, demonstrating the robustness of the EIF framework to low-fidelity side information. More detailed sensitivity analysis, including the corresponding experimental figures and specific parameter configurations, is deferred to Appendix B.4.

## 5.2. Real-World Experiments

### 5.2.1. DATASETS AND EXPERIMENTAL SETTINGS

**Dataset.** We evaluate our method on three commonly used benchmarks that span different reasoning domains: **GPQA Diamond** (Rein et al., 2024), a graduate-level science benchmark containing 198 multiple-choice questions across physics, chemistry, and biology designed to be challenging even for domain experts; **AIME 2025**, which consists of 30 problems from the American Invitational Mathematics Examination requiring advanced mathematical reasoning; and **GSM8K** (Cobbe et al., 2021), a dataset of 1319 grade school math word problems requiring multi-step arithmetic reasoning.

**Auxiliary Model Configurations.** To investigate the impact of auxiliary model selection on estimation quality, we consider two configurations for generating auxiliary labels $(W_1, W_2)$: **Config 1 (Homogeneous)**, where both auxiliary responses are generated by GPT-5 mini; and **Config 2 (Heterogeneous)**, where $W_1$ is generated by GPT-5 mini while $W_2$ is generated by DeepSeek-V3.2.

**Target Models.** Our evaluation covers 5 open-source LLMs and 5 closed-source LLMs representing different tiers of reasoning ability. These are classified into two categories: *general-purpose instruction-tuned LLMs* and *reasoning-specialized LLMs* (i.e., models explicitly trained with reinforcement learning or distillation for extended chain-of-thought reasoning). Detailed specifications for each target model are provided in Appendix C.1.

**Data Construction.** For each benchmark instance with input $X$ and ground-truth answer $G$, we construct the observation tuple $(X, Y, G, W_1, W_2, V)$ through a sequential process. First, we generate auxiliary responses $W_1$ and $W_2$ from the designated auxiliary models. Next, we obtain the target model's answer $Y$ via direct generation. Finally, we elicit the preference judgment $V \in \{0, 1\}$ by prompting the target model to compare $W_1$ and $W_2$, where $V = 1$ indicates a preference for $W_1$ and $V = 0$ for $W_2$.

We employ the standard indicator function to define the accuracy metric: $\phi_{\mathrm{acc}}(y, g) = \mathbb{1}\{y = g\}$. The target parameter of interest, $\theta = \mathbb{E}[\phi_{\mathrm{acc}}(Y, G)]$, thus represents the expected accuracy of the model across the benchmark distribution.

### 5.2.2. EXPERIMENT RESULTS AND ANALYSIS

We implement Algorithm 2 (in Appendix C.3) by employing Gemini-3-flash-preview as the outcome regression function $\widehat{\tau}(z, x)$. The marginal projection $\widehat{m}(x)$ is approximated via Monte Carlo integration using $M = 10$ auxiliary samples per instance. All the results for comparing naive estimator and the one-step estimator are shown in the Tables 1, 2, and 3. In these tables, **GT%** denotes the accuracy computed via the naive estimator (2.1) on the full dataset, serving as the ground truth. Both the naive estimator and the one-step estimator are computed on a smaller subset, with sample size $N$ indicated in each table caption, and their accuracy are reported in **Naive%** and **One-step%** respectively. **Improv.** denotes the reduction in absolute error, defined as $|\text{Naive\%} - \text{GT\%}| - |\text{One-step\%} - \text{GT\%}|$; positive values indicate that our EIF-based one-step estimator yields estimates closer to the ground truth than the naive estimator.

As demonstrated in Tables 1, 2 and 3, the one-step estimator consistently yields estimates that are significantly closer to the ground truth than those of the naive estimator, despite being restricted to the same sample budget. For instance, in the Table 2, the naive estimator for DeepSeek-V3.2(Thinking) underestimates accuracy by 3.33 percentage points (86.67% vs. 90.00% ground truth), whereas the one-step estimator reduces this bias to within 1 percentage point (91.00% and 90.10% under Config 1 and 2, respectively), yielding a substantially more calibrated approximation. This superior proximity provides empirical evidence of the enhanced sample efficiency afforded by our approach.

Furthermore, we evaluated our method across two distinct configurations for generating auxiliary data; notably, both setups yielded comparable performance gains. This consistency suggests that the EIF framework is remarkably general to variations in the auxiliary data-generating process, maintaining its advantage as long as the underlying conditional dependence is preserved.

Beyond these primary results, we further investigate the scaling behavior of the EIF estimator and its impact on performance benchmarking. Specifically, we analyze how increasing sample size enhances estimation stability and

| Model | GT% | Naive% | Config 1 | | Config 2 | |
|---|---|---|---|---|---|---|
| | | | One-step% | Improv. | One-step% | Improv. |
| ✦ Gemini-3-Flash-Preview | 90.40 | 92.00 | 91.20 | +0.80% | 89.80 | +0.99% |
| ⑨ GPT-5.2 | 86.36 | 84.00 | 85.60 | +1.60% | 85.60 | +1.60% |
| 🐋 DeepSeek-V3.2(Thinking) | 84.85 | 94.00 | 88.80 | +5.20% | 90.80 | +3.20% |
| ⊠ Grok-4-1-Fast-Reasoning | 83.33 | 88.00 | 85.80 | +2.20% | 83.60 | +4.40% |
| ✳ Claude-Sonnet-4.5 | 82.83 | 78.00 | 80.60 | +2.60% | 81.40 | +3.40% |
| 🜁 Qwen3-Next-80B-A3B-Instruct | 76.26 | 82.00 | 76.80 | +5.20% | 78.40 | +3.60% |
| 🐋 DeepSeek-R1-Distill-Llama-70B | 59.09 | 60.00 | 59.00 | +0.82% | 59.00 | +0.82% |
| 🜁 QwQ-32B-Preview | 56.06 | 60.00 | 55.40 | +3.28% | 56.00 | +3.88% |
| ∞ Llama-3.3-70B-Instruct | 46.97 | 32.00 | 37.60 | +5.60% | 37.20 | +5.20% |
| 🐋 DeepSeek-R1-Distill-Qwen-32B | 43.43 | 40.00 | 41.40 | +1.40% | 43.00 | +3.00% |

*Table 1.* GPQA Diamond EIF performance results with $N = 50$ samples (full dataset: $N = 198$). Config 1 uses GPT-5 mini for both auxiliary models; Config 2 uses GPT-5 mini and DeepSeek-V3.2. Top 3 accuracies are highlighted: 1st , 2nd , 3rd .

| Model | GT% | Naive% | Config 1 | | Config 2 | |
|---|---|---|---|---|---|---|
| | | | One-step% | Improv. | One-step% | Improv. |
| ⑨ GPT-5.2 | 96.67 | 93.33 | 96.93 | +3.08% | 96.43 | +3.10% |
| ✦ Gemini-3-Flash-Preview | 96.67 | 93.33 | 99.93 | +0.08% | 97.77 | +2.24% |
| 🐋 DeepSeek-V3.2(Thinking) | 90.00 | 86.67 | 91.00 | +2.33% | 90.10 | +3.23% |
| ⊠ Grok-4-1-Fast-Reasoning | 86.67 | 80.00 | 87.33 | +6.01% | 89.01 | +4.33% |
| ✳ Claude-Sonnet-4.5 | 83.33 | 66.67 | 70.67 | +4.00% | 78.67 | +12.00% |
| 🜁 Qwen3-Next-80B-A3B-Instruct | 73.33 | 73.33 | 72.27 | -1.06% | 78.66 | -5.33% |
| 🐋 DeepSeek-R1-Distill-Llama-70B | 53.33 | 46.67 | 50.60 | +3.93% | 54.10 | +5.89% |
| 🐋 DeepSeek-R1-Distill-Qwen-32B | 53.33 | 46.67 | 50.27 | +3.60% | 54.86 | +5.13% |
| 🜁 QwQ-32B-Preview | 30.00 | 33.33 | 29.67 | +3.00% | 29.37 | +2.70% |
| ∞ Llama-3.3-70B-Instruct | 6.67 | 6.67 | 8.33 | -1.66% | 6.24 | -0.43% |

*Table 2.* AIME 2025 EIF performance results with $N = 15$ samples (full dataset: $N = 30$). Config 1 uses GPT-5 mini for both auxiliary models; Config 2 uses GPT-5 mini and DeepSeek-V3.2. Top 3 accuracies in GT% column are highlighted: 1st , 2nd , 3rd .

| Model | GT% | Naive% | Config 1 | | Config 2 | |
|---|---|---|---|---|---|---|
| | | | One-step% | Improv. | One-step% | Improv. |
| ✦ Gemini-3-Flash-Preview | 97.88 | 98.00 | 97.90 | +0.10% | 98.30 | -0.30% |
| ⑨ GPT-5.2 | 97.50 | 97.00 | 97.65 | +0.35% | 97.00 | +0.50% |
| ✳ Claude-Sonnet-4.5 | 97.50 | 95.00 | 96.70 | +1.70% | 98.15 | +1.85% |
| ∞ Llama-3.3-70B-Instruct | 96.44 | 94.00 | 96.00 | +2.00% | 95.70 | +1.70% |
| 🐋 DeepSeek-R1-Distill-Llama-70B | 95.83 | 92.00 | 95.50 | +3.50% | 94.50 | +2.50% |
| 🜁 QwQ-32B-Preview | 95.45 | 92.00 | 93.85 | +3.40% | 94.35 | +2.35% |
| ⊠ Grok-4-1-Fast-Reasoning | 95.07 | 90.00 | 92.65 | +2.65% | 91.20 | +1.20% |
| 🐋 DeepSeek-V3.2(Thinking) | 95.00 | 92.00 | 94.00 | +2.00% | 92.90 | +0.90% |
| 🐋 DeepSeek-R1-Distill-Qwen-32B | 93.71 | 93.00 | 93.90 | +0.51% | 93.35 | +0.35% |
| 🜁 Qwen3-Next-80B-A3B-Instruct | 93.48 | 94.00 | 93.65 | +0.35% | 94.30 | +0.24% |

*Table 3.* GSM8K EIF performance results with $N = 100$ samples. Config 1 uses GPT-5 mini for both auxiliary models; Config 2 uses GPT-5 mini and DeepSeek-V3.2. Top 3 accuracies are highlighted: 1st , 2nd , 3rd .

use this refined estimator to re-rank model performance across different mathematical reasoning tasks. These extended analyses, which demonstrate the estimator's superior discriminative resolution and robustness, are detailed in Appendix C.5 and C.6.

*Remark* 5.1. We shall notice that finite-sample degradation may occur when $Z$ is weak, since the efficiency gain depends on the informativeness of the auxiliary signal. Our guarantees concern variance reduction rather than pointwise dominance. A lower variance does not imply uniformly smaller error for every realization; rather, it ensures improved performance on average.

### 5.3. Additional Robustness Experiments

We further conduct a broader set of experiments that explicitly vary all four components of the evaluator-side design:

- Auxiliary models: we add a new setting where $W_1$ and $W_2$ are generated by GPT-5 mini and QwQ-32B, respectively. The results are summarized in Table 4.

- Resampling policy: we repeat the analysis on a resampled subset of 15 problems drawn from the full AIME 2025 dataset using a different random seed. The results are summarized in Table 5.

- Selection protocol: we vary the subset size by considering a larger subset of 25 problems drawn from the full AIME dataset to examine the method's stability with respect to subset size. See Table 7.

- Prompts: we add a new correctness-only rubric (see more details in Appendix C.4 and Table 8).

| Model | Naive% | One-step% | Improve% |
|---|---|---|---|
| ✦ Gemini-3-Flash-Preview | 93.33 | 96.68 | +3.32 |
| 🐋 DeepSeek-V3.2(Thinking) | 86.67 | 87.72 | +1.05 |
| ⓧ Grok-4-1-Fast-Reasoning | 80.00 | 86.21 | +6.22 |
| ⑨ GPT-5.2 | 93.33 | 97.51 | +2.48 |
| ✳ Claude-Sonnet-4.5 | 66.67 | 72.95 | +6.29 |

*Table 4.* AIME2025 robustness under an alternative auxiliary-model pipeline, where $W_1$ and $W_2$ are generated by GPT5mini and QwQ-32B.

| Model | Naive% | One-step% | Improve% |
|---|---|---|---|
| ✳ Claude-Sonnet-4.5 | 80.00 | 82.67 | +2.66 |
| 🐋 DeepSeek-V3.2(Thinking) | 86.67 | 91.33 | +2.00 |
| ✦ Gemini-3-Flash-Preview | 100.00 | 98.00 | +2.00 |
| ⑨ GPT-5.2 | 93.33 | 98.67 | +1.33 |
| ⓧ Grok-4-1-Fast-Reasoning | 93.33 | 84.00 | +4.00 |

*Table 5.* AIME2025 robustness under a different random seed for selecting the $N = 15$ subset, using GPT-5 mini and DeepSeek-V3.2 for auxiliary generation.

Across these four checks (Tables 4 and 5, and Appendix Tables 7 and 8), the one-step estimator generally remains closer to the ground truth than the naive estimator. These results suggest that the gains are not driven by a cherry-picked auxiliary pipeline, subset, seed, or judging rubric, but persist under perturbations of the evaluation protocol.

**Equal-budget repeated decoding.** To compare with repeated decoding, we conduct an equal-budget experiment on GPQA Diamond. The one-step estimator uses one target response and 11 comparisons, while the repeated-$Y$ baseline uses 12 target responses. The one-step estimator's results under Config 1 and Config 2 are reported in Table 1; Table 6 reports the repeated-$Y$ estimates and their gaps to the ground truth.

| Model | 12-Y% (Gap to GT) |
|---|---|
| ✦ Gemini-3-Flash-Preview | 88.50 (1.90) |
| ⑨ GPT-5.2 | 85.33 (1.03) |
| 🐋 DeepSeek-V3.2(Thinking) | 83.17 (1.68) |
| ⓧ Grok-4-1-Fast-Reasoning | 79.50 (3.83) |
| ✳ Claude-Sonnet-4.5 | 76.83 (6.00) |

*Table 6.* GPQA Diamond equal-budget repeated-$Y$ baseline.

Under the same budget, the one-step estimator is closer to the ground truth than repeated-$Y$ in 4 out of 5 models for both Config 1 and Config 2, demonstrating better sample efficiency.

## 6. Conclusion

We present a semiparametric framework for evaluating the mathematical reasoning capabilities of large language models. Our approach combines direct model outputs with auxiliary pairwise preference signals to achieve statistically more efficient estimation. It is worth noting that our framework improves the statistical efficiency of estimating a fixed benchmark metric, but it does not address benchmark validity itself. If a benchmark does not faithfully measure the intended reasoning ability, a more precise estimator cannot resolve that limitation. Our method is therefore complementary to careful benchmark design and validation. Theoretically, our method attains the semiparametric efficiency bound with provable variance reduction; empirically, experiments on GPQA Diamond, AIME 2025, and GSM8K confirm more accurate estimates and reliable model rankings, especially in small-sample regimes. The framework exhibits considerable generality along two dimensions. First, the auxiliary information is not restricted to pairwise comparisons; the methodology accommodates arbitrary auxiliary signals, provided they share mutual information with the target outcome, thereby improving estimation of the target parameter. Second, the framework extends naturally to other performance metrics expressible as statistical functionals, such as pass@$k$ accuracy. Deriving the corresponding efficient influence function for such metrics to enable principled bias-corrected inference constitutes a promising direction for future work.

## Impact Statement

This paper advances efficient AI evaluation by leveraging auxiliary signals to reduce reliance on costly ground-truth annotations. This may democratize rigorous model assessment for resource-constrained researchers. However, practitioners should ensure auxiliary data sources do not introduce systematic biases that could propagate through the estimation procedure.

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

9ppSGIfpzq.

# A. Omitted Proofs

**Notation.** Throughout this appendix, for any dataset $\mathcal{D}$ and function $f$, we follow the convention and denote the sample average of $f$ on $\mathcal{D}$ and its expectation as $\mathbb{P}_{\mathcal{D}}[f] = \frac{1}{|\mathcal{D}|} \sum_{x_i \in \mathcal{D}} f(x_i), \mathbb{P}[f] = \mathbb{E}[f(X)]$. We denote the full observed data vector for a single instance as $O = (X, Y, G, Z)$, where $X$ is the input prompt, $Y$ is the model response, $G$ is the ground truth, and $Z = (W_1, W_2, V)$ represents the set of auxiliary labels. We define the metric of interest (e.g., accuracy) for a single instance as the random variable $A = \phi(Y, G)$. Consequently, our target parameter is the population mean $\theta = \mathbb{E}[A]$.

**Preliminaries.** Let $O_1, \ldots, O_N \overset{\text{i.i.d.}}{\sim} P \in \mathcal{M}$, where $\mathcal{M}$ is the semiparametric model space in Definition 3.1, and suppose each element of $\mathcal{M}$ admits a density with respect to a common dominating measure $\mu$. Consider a one-dimensional regular parametric submodel $\{P_t : t \in (-\varepsilon, \varepsilon)\} \subset \mathcal{M}$ with $P_0 = P$; its *score function* at $t = 0$ is $\dot{\ell}(O) = \partial \log p_t(O)/\partial t \big|_{t=0} \in L_2^0(P)$, where $L_2^0(P) = \{h \in L_2(P) : \mathbb{E}[h] = 0\}$. The *tangent set* at $P$ is the collection of all such scores over all regular parametric submodels of $\mathcal{M}$; the *tangent space* $\mathcal{S}$ is its closed linear span in $L_2^0(P)$:

$$\mathcal{S} = \overline{\text{lin}}\left\{ \dot{\ell} \in L_2^0(P) : \dot{\ell} \text{ is the score of some regular parametric submodel of } \mathcal{M} \text{ at } P \right\}.$$

We use lowercase $p$ and $p_t$ to denote densities of $P$ and $P_t$ with respect to the dominating measure $\mu$. For additional background on these semiparametric concepts, see van der Vaart (1998, Chapter 25).

## A.1. Proof of Proposition 3.3

**Proof sketch.** The idea is: the most efficient estimator is constructed based on its EIF constrained to the corresponding tangent space. Since $p(Z \mid X)$ is fixed, the tangent space decomposes orthogonally into variation in the marginal of $X$ and variation in the conditional of $(Y, G)$ given $(X, Z)$. The EIF is obtained by projecting the nonparametric influence function $\phi(Y, G) - \theta$ onto these two components: the conditional projection yields the residual $\phi(Y, G) - \tau(X, Z)$, and the marginal projection yields $m(X) - \theta$. Summing them gives the EIF in Equation (3.1).

*Proof.* To derive the Efficient Influence Function (EIF), we project the canonical gradient onto the tangent space of the constrained semiparametric model. The joint density factorizes as $p(O) = p(y, g \mid Z, x)p(Z \mid x)p(x)$. Crucially, $p(Z \mid x)$ is known and fixed, while the other components are nonparametric.

**Step 1: Decomposition of the Tangent Space.** The tangent space $\mathcal{S}$ consists of the closure of all score functions associated with parametric submodels. Due to the factorization structure and the constraint that $p(Z \mid x)$ is fixed, $\mathcal{S}$ decomposes into two orthogonal subspaces $\mathcal{S} = \mathcal{T}_1 \oplus \mathcal{T}_2$:

$$\mathcal{T}_1 = \{s_1(X) \in L_2(P) : \mathbb{E}[s_1(X)] = 0\},$$
$$\mathcal{T}_2 = \{s_2(O) \in L_2(P) : \mathbb{E}[s_2(O) \mid X, Z] = 0\}.$$

Here, $\mathcal{T}_1$ corresponds to perturbations of the marginal $p(x)$, and $\mathcal{T}_2$ corresponds to perturbations of the conditional $p(y, g \mid Z, x)$. Scores corresponding to $p(Z \mid x)$ are zero because this distribution is fixed. We verify orthogonality by iterated expectations: for any $s_1 \in \mathcal{T}_1, s_2 \in \mathcal{T}_2$, $\mathbb{E}[s_1 s_2] = \mathbb{E}[s_1(X)\mathbb{E}[s_2(O) \mid X, Z]] = 0$.

**Step 2: The Initial Gradient.** In a fully nonparametric model (where all components are unknown), the influence function for the mean parameter $\theta$ is simply the centered random variable:

$$\psi_{\text{np}}(O) = A - \theta.$$

The EIF in our constrained model is the projection of $\psi_{\text{np}}$ onto $\mathcal{S} = \mathcal{T}_1 \oplus \mathcal{T}_2$, i.e., $\psi = \mathbb{E}[\psi_{\text{np}} \mid \mathcal{S}] = \mathbb{E}[\phi(Y, G) - \theta \mid \mathcal{S}]$. Since the subspaces are orthogonal, the projection is the sum of separate projections.

**Step 3: Projection onto $\mathcal{T}_2$.** The projection of any $U \in L_2(P)$ onto the space of conditional mean-zero functions $\mathcal{T}_2$ is given by $U - \mathbb{E}[U \mid X, Z]$. Applying this to $\psi_{\text{np}}$:

$$\Pi(\psi_{\text{np}} \mid \mathcal{T}_2) = (A - \theta) - \mathbb{E}[A - \theta \mid X, Z]$$
$$= A - \mathbb{E}[A \mid X, Z].$$

Recalling the definition of the full regression function $\tau(X, Z) = \mathbb{E}[\phi(Y, G) \mid Z, X]$, this yields:

$$\Pi(\psi_{\text{np}} \mid \mathcal{T}_2) = \phi(Y, G) - \tau(X, Z).$$

**Step 4: Projection onto $\mathcal{T}_1$.** The projection of any $U \in L_2(P)$ onto the space of marginal mean-zero functions $\mathcal{T}_1$ is given by $\mathbb{E}[U \mid X] - \mathbb{E}[U]$. Applying this to $\psi_{\text{np}}$:

$$\Pi(\psi_{\text{np}} \mid \mathcal{T}_1) = \mathbb{E}[A - \theta \mid X] - \underbrace{\mathbb{E}[A - \theta]}_{=0} = \mathbb{E}[A \mid X] - \theta.$$

To evaluate $\mathbb{E}[A \mid X]$, we apply the law of iterated expectations conditioning on $Z$. Since $p(Z \mid x)$ is known, this expectation is equivalent to the integrated regression function $m(X)$:

$$\mathbb{E}[A \mid X] = \mathbb{E}_{Z|X}\Big[\mathbb{E}[A \mid X, Z]\Big] = \int \tau(X, Z)\, dP(Z \mid X) = m(X).$$

Thus, the projection is:

$$\Pi(\psi_{\text{np}} \mid \mathcal{T}_1) = m(X) - \theta.$$

**Step 5: Summation.** The Efficient Influence Function is the sum of the projections derived in Steps 3 and 4:

$$\psi(O) = \Big(\phi(Y, G) - \tau(X, Z)\Big) + \Big(m(X) - \theta\Big).$$

$\square$

## A.2. Proof of Theorem 4.5

*Proof.* Let $K$ be the number of folds. Let $I_k$ denote the indices of the $k$-th fold, and $I_k^c$ denote the complement (training set). Let $\widehat{\tau}^{(-k)}$ and $\widehat{m}^{(-k)}$ be the estimators trained on $I_k^c$. The estimated influence function for data point $i \in I_k$ is:

$$\widehat{\psi}_i = \widehat{m}^{(-k)}(X_i) + \phi(Y_i, G_i) - \widehat{\tau}^{(-k)}(Z_i, X_i). \tag{A.1}$$

The true influence function is $\psi_i = m(X_i) + \phi(Y_i, G_i) - \tau(Z_i, X_i) - \theta$.

We decompose the empirical process for the estimator $\widehat{\theta} = \frac{1}{N} \sum_{k=1}^{K} \sum_{i \in I_k} \widehat{\psi}_i$:

$$\sqrt{N}(\widehat{\theta} - \theta) = \frac{1}{\sqrt{N}} \sum_{k=1}^{K} \sum_{i \in I_k} \widehat{\psi}_i - \sqrt{N}\theta$$

$$= \frac{1}{\sqrt{N}} \sum_{i=1}^{N} \psi_i \quad \text{(Main Term)} \tag{A.2}$$

$$+ \frac{1}{\sqrt{N}} \sum_{k=1}^{K} \sum_{i \in I_k} (\widehat{\psi}_i - \psi_i). \quad \text{(Remainder Term)} \tag{A.3}$$

**Step 1: The Main Term.** Term (A.2) is $\sqrt{N}\mathbb{P}_{\mathcal{D}}[\psi]$. Under Assumption 4.1, we have $\mathbb{E}[\phi^2] < \infty$. By Jensen's inequality, $\mathbb{E}[\tau^2] \leq \mathbb{E}[\phi^2] < \infty$, which implies that the influence function $\psi$ has a finite variance (i.e., $\text{Var}(\psi) < \infty$). Since the observations are i.i.d. and $\mathbb{P}[\psi] = 0$, by the Central Limit Theorem, this term converges to $\mathcal{N}(0, \sigma_{\text{eff}}^2)$.

**Step 2: Analysis of the Remainder Term.** Let $R_N$ denote the remainder term (A.3). We define the difference function for fold $k$ as $\Delta^{(-k)}(O) = \widehat{\psi}^{(-k)}(O) - \psi(O)$. We can decompose the sum over fold $I_k$ into an empirical process term and a bias term:

$$\frac{1}{\sqrt{N}} \sum_{i \in I_k} \Delta^{(-k)}(O_i) = \sqrt{\frac{N_k}{N}} \sqrt{N_k}\Big(\mathbb{P}_{I_k}[\Delta^{(-k)}] - \mathbb{P}[\Delta^{(-k)}]\Big) + \sqrt{\frac{N_k}{N}} \sqrt{N_k}\mathbb{P}[\Delta^{(-k)}]. \tag{A.4}$$

**Step 2a: The Stochastic Term.** Conditional on the training data $I_k^c$, we compute the variance of the scaled stochastic term. Since the summands are i.i.d. given $I_k^c$, we have:

$$\text{Var}\Big(\sqrt{N_k}\Big(\mathbb{P}_{I_k}[\Delta^{(-k)}] - \mathbb{P}[\Delta^{(-k)}]\Big) \,\Big|\, I_k^c\Big) = \text{Var}\Big(\frac{1}{\sqrt{N_k}} \sum_{i \in I_k} \Delta^{(-k)}(O_i) \,\Big|\, I_k^c\Big)$$

$$= \text{Var}(\Delta^{(-k)} \mid I_k^c).$$

We bound this variance using the $L_2$ estimation error:

$$\text{Var}(\Delta^{(-k)} \mid I_k^c) \leq \mathbb{P}\Big[(\Delta^{(-k)})^2 \mid I_k^c\Big]$$
$$\leq 2\|\widehat{m}^{(-k)} - m\|_{P,2}^2 + 2\|\widehat{\tau}^{(-k)} - \tau\|_{P,2}^2.$$

By Chebyshev's inequality applied to the conditional distribution, for any $\epsilon > 0$:

$$\mathbb{P}\Big(\Big|\sqrt{N_k}\Big(\mathbb{P}_{I_k}[\Delta^{(-k)}] - \mathbb{P}[\Delta^{(-k)}]\Big)\Big| > \epsilon \,\Big|\, I_k^c\Big) \leq \frac{\text{Var}(\Delta^{(-k)} \mid I_k^c)}{\epsilon^2}$$
$$\leq \frac{2\|\widehat{m}^{(-k)} - m\|_{P,2}^2 + 2\|\widehat{\tau}^{(-k)} - \tau\|_{P,2}^2}{\epsilon^2}.$$

To obtain the unconditional convergence, we take the expectation over the training data $I_k^c$. Let $E_k$ denote the upper bound derived above. We decompose the error term $\|\widehat{m}^{(-k)} - m\|_{P,2}^2$ by introducing the exact integrated regression function $\widetilde{m}^{(-k)}(X) = \mathbb{E}_{Z|X}[\widehat{\tau}^{(-k)}(Z, X)]$. Using the inequality $(a + b)^2 \leq 2a^2 + 2b^2$:

$$\|\widehat{m}^{(-k)} - m\|_{P,2}^2 = \|\widehat{m}^{(-k)} - \widetilde{m}^{(-k)} + \widetilde{m}^{(-k)} - m\|_{P,2}^2$$
$$\leq 2\underbrace{\|\widehat{m}^{(-k)} - \widetilde{m}^{(-k)}\|_{P,2}^2}_{\text{Monte Carlo Error}} + 2\underbrace{\|\widetilde{m}^{(-k)} - m\|_{P,2}^2}_{\text{Statistical Error}}.$$

The statistical error $\|\widetilde{m}^{(-k)} - m\|_{P,2}^2$ is bounded by $\|\widehat{\tau}^{(-k)} - \tau\|_{P,2}^2$ via Jensen's inequality. The Monte Carlo error corresponds to the variance of the integration approximation, which is $O_p(1/M)$. Substituting this decomposition back into the expression for $E_k$:

$$E_k \leq \frac{4\|\widehat{m}^{(-k)} - \widetilde{m}^{(-k)}\|_{P,2}^2 + 6\|\widehat{\tau}^{(-k)} - \tau\|_{P,2}^2}{\epsilon^2}.$$

Under Assumption 4.2 (consistency of $\widehat{\tau}$) and Assumption 4.3 (sufficient Monte Carlo samples, $M \to \infty$), both terms in the numerator converge to zero in probability.

Since the conditional probability is bounded by 1, we use the bounded convergence argument:

$$\mathbb{P}\Big(\Big|\sqrt{N_k}\Big(\mathbb{P}_{I_k}[\Delta^{(-k)}] - \mathbb{P}[\Delta^{(-k)}]\Big)\Big| > \epsilon\Big) = \mathbb{E}\Big[P\Big(\Big|\sqrt{N_k}\Big(\mathbb{P}_{I_k}[\Delta^{(-k)}] - \mathbb{P}[\Delta^{(-k)}]\Big)\Big| > \epsilon \,\Big|\, I_k^c\Big)\Big]$$
$$\leq \mathbb{E}[\min(1, E_k)].$$

Since $E_k \xrightarrow{p} 0$ and the function $f(x) = \min(1, x)$ is bounded and continuous at 0, it follows that $\mathbb{E}[\min(1, E_k)] \to 0$. Thus, the stochastic term is $o_p(1)$.

**Step 2b: The Bias Term.** We examine the unconditional expectation of the difference function $\mathbb{P}[\Delta^{(-k)}]$. By the Law of Iterated Expectations, $\mathbb{P}[\Delta^{(-k)}] = \mathbb{E}_{I_k^c}[\mathbb{E}[\Delta^{(-k)} \mid I_k^c]]$. We analyze the inner conditional expectation.

Let $\widetilde{m}^{(-k)}(X) = \mathbb{E}_{Z|X}[\widehat{\tau}^{(-k)}(X, Z) \mid X]$ denote the *exact* integrated regression function. We decompose the bias into the Monte Carlo approximation error and the structural error:

$$\mathbb{E}[\Delta^{(-k)} \mid I_k^c] = \mathbb{E}\Big[(\widehat{m}^{(-k)} - m) - (\widehat{\tau}^{(-k)} - \tau) \mid I_k^c\Big]$$
$$= \mathbb{E}\Big[(\widehat{m}^{(-k)} - \widetilde{m}^{(-k)}) + (\widetilde{m}^{(-k)} - \widehat{\tau}^{(-k)}) \mid I_k^c\Big] - \underbrace{\mathbb{E}[m - \tau]}_{=0}. \tag{A.5}$$

First, we analyze the Monte Carlo term. Since $\widehat{m}^{(-k)}(X)$ is constructed by averaging $\widehat{\tau}^{(-k)}$ over $M$ i.i.d. samples drawn from $P(Z|X)$, it is an unbiased estimator of $\widetilde{m}^{(-k)}(X)$. Thus:

$$\mathbb{E}_{Z_{1:M}|X}\Big[\widehat{m}^{(-k)}(X)\Big] = \widetilde{m}^{(-k)}(X) \implies \mathbb{E}\Big[\widehat{m}^{(-k)} - \widetilde{m}^{(-k)} \mid I_k^c\Big] = 0. \tag{A.6}$$

Second, we analyze the structural term involving $\widehat{\tau}^{(-k)}$. Applying the Law of Iterated Expectations:

$$\mathbb{E}_{X,Z}\Big[\widehat{\tau}^{(-k)}(X, Z)\Big] = \mathbb{E}_X\Big[\mathbb{E}_{Z|X}[\widehat{\tau}^{(-k)}(X, Z) \mid X]\Big] = \mathbb{E}_X\Big[\widetilde{m}^{(-k)}(X)\Big]. \tag{A.7}$$

Therefore, $\mathbb{E}\big[\widetilde{m}^{(-k)} - \widehat{\tau}^{(-k)} \mid I_k^c\big] = 0$.

Consequently, $\mathbb{E}[\Delta^{(-k)} \mid I_k^c] = 0$. The bias term vanishes exactly. This property holds because the Monte Carlo simulation is unbiased, and the relationship between $\widetilde{m}$ and $\widehat{\tau}$ is enforced structurally via the known distribution $P(Z|X)$.

**Conclusion.** Since $R_N = o_p(1)$, we have $\sqrt{N}(\widehat{\theta} - \theta) = \frac{1}{\sqrt{N}} \sum \psi_i + o_p(1)$, which converges to $\mathcal{N}(0, \sigma_{\text{eff}}^2)$. $\qquad\square$

## A.3. Proof of Corollary 4.7

*Proof.* To prove the strict variance reduction, we utilize the Law of Total Variance to decompose the variance of the naive estimator.

Recall the definitions of the regression functions:

$$\tau(X, Z) = \mathbb{E}[\phi(Y, G) \mid Z, X],$$
$$m(X) = \mathbb{E}[\phi(Y, G) \mid X] = \mathbb{E}[\tau(X, Z) \mid X].$$

We decompose the total variance $\sigma_{\text{naive}}^2 = \text{Var}(\phi(Y, G))$ by iteratively conditioning on $X$ and then on $(X, Z)$:

$$\text{Var}(\phi(Y, G)) = \text{Var}\Big(\mathbb{E}[\phi(Y, G) \mid X]\Big) + \mathbb{E}\Big[\text{Var}(\phi(Y, G) \mid X)\Big]$$

$$= \text{Var}(m(X)) + \mathbb{E}\Big[\underbrace{\text{Var}\big(\mathbb{E}[\phi(Y, G) \mid Z, X] \mid X\big)}_{\text{Variance explained by } Z} + \text{Var}\big(\phi(Y, G) \mid Z, X\big)\Big]$$

$$= \text{Var}(m(X)) + \mathbb{E}\Big[\text{Var}(\tau(X, Z) \mid X)\Big] + \mathbb{E}\Big[\text{Var}(\phi(Y, G) \mid Z, X)\Big]. \tag{A.8}$$

Now consider the variance of our efficient estimator. The Efficient Influence Function is $\psi = (\phi - \tau) + (m - \theta)$. Since the residual $(\phi - \tau)$ is orthogonal to any function of $X$ (and thus orthogonal to $m(X)$), the variance is:

$$\sigma_{\text{eff}}^2 = \text{Var}(\psi) = \text{Var}(m(X)) + \text{Var}\big(\phi(Y, G) - \tau(X, Z)\big)$$

$$= \text{Var}(m(X)) + \mathbb{E}\Big[\text{Var}(\phi(Y, G) \mid Z, X)\Big]. \tag{A.9}$$

Subtracting (A.9) from (A.8), the difference is exactly the middle term:

$$\sigma_{\text{naive}}^2 - \sigma_{\text{eff}}^2 = \mathbb{E}\Big[\text{Var}\big(\tau(X, Z) \mid X\big)\Big]. \tag{A.10}$$

To rigorously establish the strict inequality $\sigma_{\text{eff}}^2 < \sigma_{\text{naive}}^2$, we proceed by contradiction.

First, recall the variance decomposition relation:

$$\sigma_{\text{naive}}^2 = \sigma_{\text{eff}}^2 + \mathbb{E}\left[\text{Var}(\tau(X, Z) \mid X)\right] = \sigma_{\text{eff}}^2 + \mathbb{E}\left[\big(\tau(X, Z) - m(X)\big)^2\right].$$

Suppose, for the sake of contradiction, that there is no efficiency gain, i.e., $\sigma_{\text{eff}}^2 = \sigma_{\text{naive}}^2$. Substituting this into the decomposition implies:

$$\mathbb{E}\left[\big(\tau(X, Z) - m(X)\big)^2\right] = 0.$$

Since the squared term is non-negative, its expectation being zero implies that the term itself must be zero almost surely:

$$\tau(X, Z) = m(X) \quad \text{a.s.}$$

This equality asserts that the auxiliary variable $Z$ is conditionally mean-independent of the target metric given $X$. However, this contradicts our premise that $Z$ contains non-redundant information, specifically the condition that:

$$\tau(X, Z) \neq m(X)$$

on a set of positive probability. Therefore, the assumption $\sigma_{\text{eff}}^2 = \sigma_{\text{naive}}^2$ must be false, and we conclude that the inequality is strict:

$$\sigma_{\text{eff}}^2 < \sigma_{\text{naive}}^2.$$

□

## A.4. Orthogonality of EIF Components

We show that the two additive terms in the EIF, $(m(X) - \theta)$ and $(\phi(Y, G) - \tau(X, Z))$, are orthogonal (i.e., have zero covariance).

*Proof.* First, note that both terms have mean zero:

$$\mathbb{E}[m(X) - \theta] = \mathbb{E}[m(X)] - \theta = \theta - \theta = 0,$$
$$\mathbb{E}[\phi(Y, G) - \tau(X, Z)] = \mathbb{E}[\phi(Y, G)] - \mathbb{E}[\tau(X, Z)] = \theta - \theta = 0.$$

To show orthogonality, we compute:

$$\mathrm{Cov}\big(m(X) - \theta, \phi(Y, G) - \tau(X, Z)\big)$$
$$= \mathbb{E}\Big[(m(X) - \theta)\big(\phi(Y, G) - \tau(X, Z)\big)\Big]$$
$$= \mathbb{E}\Big[\mathbb{E}\big[(m(X) - \theta)(\phi(Y, G) - \tau(X, Z)) \mid X, Z\big]\Big]$$
$$= \mathbb{E}\Big[(m(X) - \theta) \cdot \mathbb{E}\big[\phi(Y, G) - \tau(X, Z) \mid X, Z\big]\Big].$$

By the definition of $\tau(X, Z) = \mathbb{E}[\phi(Y, G) \mid X, Z]$, we have:

$$\mathbb{E}[\phi(Y, G) - \tau(X, Z) \mid X, Z] = \tau(X, Z) - \tau(X, Z) = 0.$$

Therefore:

$$\mathrm{Cov}\big(m(X) - \theta, \phi(Y, G) - \tau(X, Z)\big) = \mathbb{E}\Big[(m(X) - \theta) \cdot 0\Big] = 0.$$

This orthogonality implies that the variance of the EIF decomposes additively:

$$\mathrm{Var}(\psi) = \mathrm{Var}(m(X) - \theta) + \mathrm{Var}(\phi(Y, G) - \tau(X, Z)).$$

□

# B. Details of Simulation Study

## B.1. Derivation of the Efficient Influence Function

We explicitly derive the Efficient Influence Function (EIF) $\psi(O)$ for this specific setting. Recall the general form from equation (3.1):

$$\psi(O) = \Big(m(X) - \theta\Big) + \Big(\phi(Y, G) - \tau(X, Z)\Big)$$

**The Integrated Regression Function $m(X)$.**    Since the noise $\epsilon_{li}$ is independent of the input $X$, the conditional expectation of the metric function $\phi(Y, G)$ given only $X$ is constant:

$$m(X) = \mathbb{E}[\epsilon_{li}^2 \mid X] = \mathbb{E}[\epsilon_{li}^2] = \sigma_l^2 = \theta_l.$$

Thus, the term $(m(X) - \theta_l)$ vanishes exactly.

**The Full Regression Function $\tau(X, Z)$.**    Let $S = W - X = \rho\epsilon + \eta$ be the observed signal centered at zero for simplicity. Since the noise $\eta$ is normal with zero mean, the joint distribution of $(\epsilon, S)$ is therefore multivariate normal with zero mean. By the property of conditional mean of multivariate normal, we obtain the conditional expectation of $\epsilon$ given $S$ to be linear:

$$\mathbb{E}[\epsilon \mid S] = \kappa S, \text{ where } \kappa = \frac{\mathrm{Cov}(\epsilon, S)}{\mathrm{Var}(S)} = \frac{\rho\sigma_l^2}{\rho^2\sigma_l^2 + \sigma_\eta^2}.$$

By the law of total variance that $\mathbb{E}[\epsilon^2 \mid S] = \mathrm{Var}(\epsilon \mid S) + (\mathbb{E}[\epsilon \mid S])^2$, the regression function $\tau(X, Z)$ is simplified to:

$$\tau(X, Z) = \mathbb{E}[\epsilon^2 \mid S] = \underbrace{(1 - \kappa\rho)\sigma_l^2}_{\text{Intercept}} + \underbrace{\kappa^2(W - X)^2}_{\text{Quadratic}}. \tag{B.1}$$

This formula highlights a critical requirement for implementation in this setting that the regression model must incorporate quadratic additional features to capture the relationship between the linear auxiliary variable $W$ and the quadratic metric $\phi$.

### B.2. Theoretical Variance Reduction

As established in Section 4.2, the one-step estimator is theoretically guaranteed to yield significant efficiency gains. To bridge the gap between theory and the observed ranking performance, we provide a formal variance reduction analysis specific to this simulation setting. This is accompanied by numerical simulations that offer a more intuitive visualization of the variance reduction, confirming that the superior ranking accuracy is a direct manifestation of the estimator's reduced uncertainty.

**Variance of the Naive Estimator.** The naive estimator is based on the function $\phi(Y, G) = (Y - G)^2 = \epsilon^2$. For the Gaussian variable $\epsilon \sim \mathcal{N}(0, \sigma_l^2)$, the variance of its square is determined by the fourth moment as follows:

$$\sigma_{\text{naive}}^2 := \mathrm{Var}(\phi) = \mathrm{Var}(\epsilon^2) = \mathbb{E}[\epsilon^4] - (\mathbb{E}[\epsilon^2])^2 = 3\sigma_l^4 - \sigma_l^4 = 2\sigma_l^4.$$

**Variance of the One-Step Estimator.** By the law of total variance, the variance of the influence function $\psi = \phi - \tau(X, Z) + m(X)$ is the residual variance after regression:

$$\sigma_{\text{1-step}}^2 := \mathrm{Var}(\psi) = \mathrm{Var}(\phi - \tau(X, Z)) = \sigma_{\text{naive}}^2 - \mathrm{Var}(\tau(X, Z)).$$

From the derivation of $\tau$ in (B.1), the explained variance comes from the term $k^2 S^2$. Since $S = \rho\epsilon + \eta$ where $\epsilon$ and $\eta$ are Gaussian random variables, we have $S \sim \mathcal{N}(0, \mathrm{Var}(S))$ to be normal, and $\mathrm{Var}(S^2) = 2[\mathrm{Var}(S)]^2$. Therefore:

$$\mathrm{Var}(\tau) = 2\left(\frac{\mathrm{Cov}(\epsilon, S)^2}{\mathrm{Var}(S)}\right)^2. \tag{B.2}$$

Let $R^2$ denote the coefficient of determination:

$$R^2 = \frac{\mathrm{Cov}(\epsilon, S)^2}{\mathrm{Var}(\epsilon)\mathrm{Var}(S)} = \frac{\rho^2\sigma_l^2}{\rho^2\sigma_l^2 + \sigma_\eta^2}.$$

Substituting this back to (B.2), we obtain $\mathrm{Var}(\tau) = 2\sigma_l^4(R^2)^2$. Therefore, we obtain the variance of the one-step estimator as follows:

$$\sigma_{\text{1-step}}^2 = 2\sigma_l^4\left[1 - (R^2)^2\right]. \tag{B.3}$$

**Simulation Results and Analysis** Based on the form of the variance of naive estimator and one-step estimator, we use the variance reduction ratio (VR) as our criterion, which is given by:

$$\mathrm{VR} = 1 - \frac{\sigma_{\text{1-step}}^2}{\sigma_{\text{naive}}^2} = (R^2)^2.$$

This result indicates that the performance of our one-step estimator is better if the VR is more close to 1.

To investigate the fundamental mechanism behind this superior ranking performance, we analyze the variance reduction (VR) achieved by our estimator. As shown in Figure 3, the Oracle VR, where the nuisance function $m(X)$ is replaced by its closed-form theoretical expression, demonstrates that the efficiency gain of our one-step estimator asymptotically approaches 1 as the model-specific signal $\sigma_l^2$ increases. Our empirical one-step VR confirms this desirable property, showing a consistent upward trend toward the Oracle limit as the base variance of $Y$ escalates. This validates the improving relative performance of our estimator in high-noise regimes; intuitively, as the intrinsic signal $\sigma_l^2$ becomes more dominant, the structural information captured from auxiliary data provides increasingly critical corrections for the final estimation.

Moreover, as illustrated in Figure 4, the Oracle VR indicates that the efficiency gain of the one-step estimator diminishes as the noise in the auxiliary information, $\sigma_\eta^2$, increases. This downward trend is expected, as higher auxiliary noise obscures

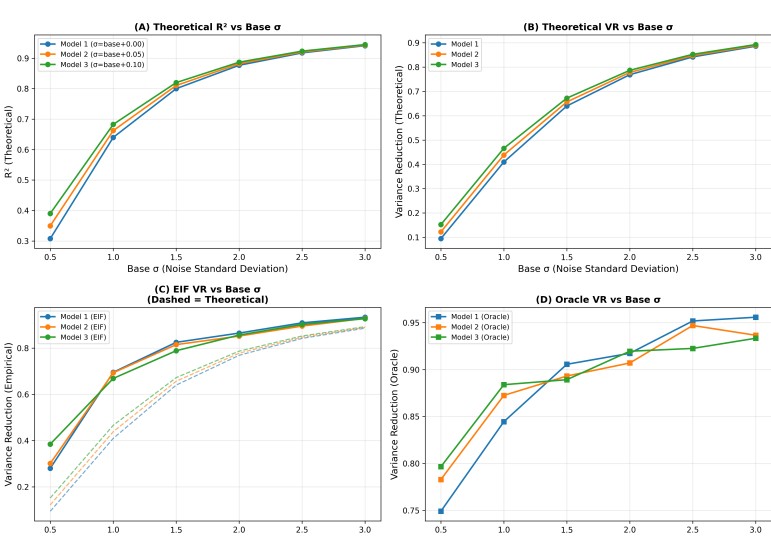

*Figure 3.* Variance reduction vs. model-specific signal

the structural signal. Our empirical EIF VR results closely follow this theoretical trajectory, confirming the estimator's sensitivity to the fidelity of auxiliary data.

### B.3. Ranking Accuracy vs. Model-specific Signal

In this section, we evaluate ranking performance in relation to the base noise level, while maintaining a constant gap of 0.05 between the model-specific variances $\sigma_l^2$ (Figure 2). It is important to note that although we refer to $\sigma_l^2$ as the "signal" parameter, it simultaneously controls both the signal strength and the noise magnitude in our simulation setup, since the metric $\phi = \epsilon_l^2$ where $\epsilon_l \sim \mathcal{N}(0, \sigma_l^2)$.

When we shift the model variances by a common base value (e.g., $\sigma_1^2 = c, \sigma_2^2 = c + 0.05, \sigma_3^2 = c + 0.1$ for varying $c$), the *relative gaps* between models remain constant—the true ranking is preserved. However, the absolute variance of the metric $\phi$ increases quadratically with $\sigma_l^2$ (since $\text{Var}(\epsilon^2) = 2\sigma_l^4$ for Gaussian $\epsilon$). This poses a significant challenge for the naive estimator: as the noise level grows, distinguishing between models with similar performance becomes increasingly difficult, leading to degraded ranking accuracy.

In contrast, the one-step estimator maintains robust ranking performance even in high-noise regimes. This resilience stems from the efficiency property of the EIF-based estimator: by leveraging auxiliary information, it achieves variance reduction that scales favorably with the noise level. Specifically, as shown in our variance analysis, the variance reduction ratio $(R^2)^2$ approaches 1 as $\sigma_l^2$ increases, meaning the one-step estimator captures an increasingly larger fraction of the total variance. Consequently, while absolute estimation uncertainty grows for both methods, the one-step estimator's relative precision advantage becomes more pronounced, preserving its ability to correctly rank models even when the naive estimator fails.

### B.4. Ranking Accuracy vs. Auxiliary Noise

In this section, we provide a detailed analysis of how the information density of the auxiliary variables influences the estimation gains by varying the intensity of the auxiliary noise $\sigma_\eta^2$. In this setup, we fix the variances of the model-specific variances at $\sigma_1^2 = 1.0, \sigma_2^2 = 1.05$, and $\sigma_3^2 = 1.1$ respectively to represent a challenging evaluation scenario. This provides a clear test of our method's reliance on high-quality side information. Figure 5 depicts the impact of auxiliary information fidelity on ranking performance. As the noise level $\sigma_\eta^2$ escalates, the predictive power of the auxiliary responses diminishes, leading to a gradual decrease in ranking accuracy for the one-step estimator. However, even in high-noise regimes, the one-step estimator effectively extracts the remaining signal to outperform the Naive baseline, which confirms that the EIF structure can leverage even weak correlations to enhance the ability of discrimination. The Naive baseline, being independent of the auxiliary signal, provides a constant reference point that underscores the efficiency gains provided by the EIF structure.

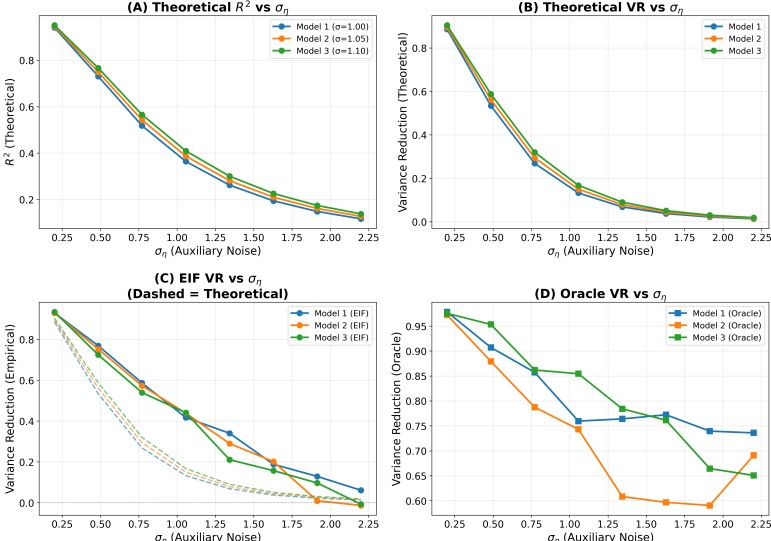

*Figure 4.* Variance reduction vs. auxiliary noise

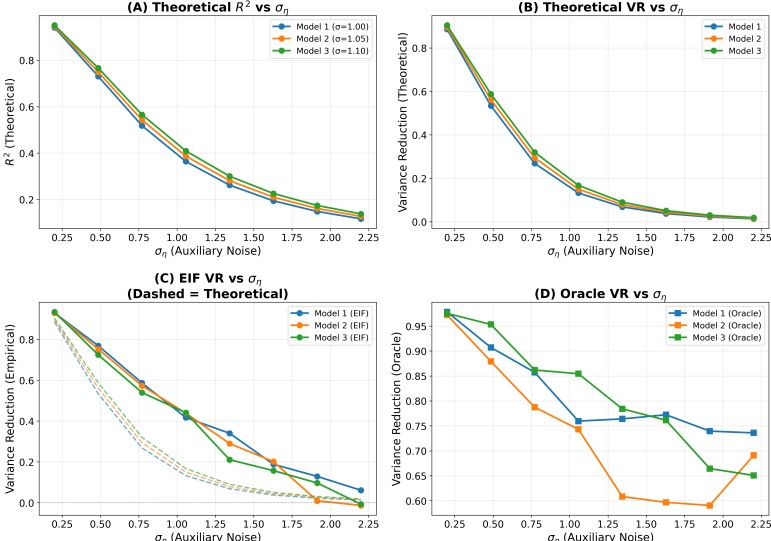

*Figure 5.* Ranking accuracy for different auxiliary noise

## C. Details of Real World Experiment

### C.1. Target LLMs

We evaluate ten state-of-the-art large language models spanning both closed-source and open-source categories, including: GPT-5.2 (OpenAI, 2025), Gemini-3-Flash-Preview (Google Research, 2025), Claude-Sonnet-4.5 (Anthropic Inc, 2025), Llama-3.3-70B-Instruct (Grattafiori et al., 2024; Meta AI, 2024), Qwen3-Next-80B-A3B-Instruct. Additionally, we conduct experiment on several o1-like LLMs, such as Grok-4-1-Fast-Reasoning (xAI, 2025), DeepSeek-V3.2(Thinking) (Liu et al., 2025a), QwQ-32B-Preview (Qwen Team, 2024), DeepSeek-R1-Distill-Llama-70B (Guo et al., 2025), DeepSeek-R1-Distill-Qwen-32B (Guo et al., 2025). This selection encompasses diverse model architectures, parameter scales, and training paradigms, enabling assessment of the generalizability of our variance reduction approach.

### C.2. Details of Model Setting

#### C.2.1. MAX TOKEN CONSTRAINT

To account for the challenging nature of the tasks and the extensive token requirements of reasoning-heavy models, we standardized the maximum token limit at 32,768 for target model outputs. This constraint was chosen to prevent premature truncation, thereby ensuring that the target model's accuracy reflects its true performance without being artificially limited by context length restrictions.

Regarding the auxiliary variables $W_1$ and $W_2$, we implement a reduced token budget (capped at 8,192 or 2,048), as the

predictive signal for our one-step estimator is primarily encoded within the intermediate reasoning traces rather than being confined to the final answer. Since the EIF framework leverages the semantic alignment between the target and auxiliary processes, the early hidden states and partial logic chains provide sufficient information to stabilize the estimation. This allows for significant computational savings without compromising the variance reduction, even if the auxiliary sequence is truncated before reaching an explicit conclusion. Furthermore, this reduction accommodates specific architectural constraints. For example, since the QwQ-32B-Preview model imposes a strict context window of 32K tokens, limiting the length of $W_1$ and $W_2$ is necessary to ensure the total input—comprising the query and both auxiliary responses—remains within the model's operational limits.

*Remark* C.1. Reasoning models in the DeepSeek family exhibit high verbosity primarily due to their exhaustive reasoning structure; they typically begin by restating the problem and detailing a comprehensive execution plan before generating the final response. Furthermore, these models frequently engage in explicit self-correction loops—interjecting phrases such as 'Wait a minute, let me double-check'—which triggers a full re-evaluation of the prior logic chain and significantly inflates token consumption. Similarly, Qwen3-Next-80B often produces excessively long outputs characterized by semantic redundancy, where the model iteratively restates its underlying logic, leading to repetitive answer structures. Therefore, appropriately reducing the maximum token limit for auxiliary data potentially refines the evaluation process. This approach retains the integrity of the model's initial reasoning path while eliminating redundant verbiage that may interfere with accurate preference assessment.

### C.3. Real Data Algorithm

Our initial approach involved transforming the textual content generated by the target models into dense embeddings using a pre-trained model, such as Qwen-embedding, and subsequently training a Multilayer Perceptron (MLP) as a regressor. However, the extensive length of the LLM-generated responses results in high-dimensional feature vectors that present significant challenges for model training. As noted in Remark 3.5, this high-dimensional setting—where the feature dimensionality far exceeds the available sample size—leads to severe computational intractability and the risk of overfitting, making it difficult to train a robust regression model with limited data.

Inspired by the advanced reasoning capabilities of Large Language Models, we reformulate the regression task as an In-Context Learning problem. Specifically, the conditional expectation $\tau(X, Z)$ can be conceptualized as a mapping where the covariates $(X, Z)$ serve as the input prompt, and the regressor output corresponds to the model-generated response. Given their inherent ability to evaluate and critique complex textual data, we leverage these reasoning models to logically derive the outcome regressor. This approach treats the regression not as a traditional curve-fitting exercise, but as a semantic inference task facilitated by the model's sophisticated understanding of the benchmark context. In our experiment, we regard the Gemini-3-Flash as the already-trained semantic regressor in the Algorithm 2.

*Remark* C.2. While both zero-shot and ICL prompting are viable for predicting $\tau_{LLM}$, we observed that Gemini 3 Flash delivers significantly more reliable outputs in the zero-shot setting. This phenomenon stems from several practical constraints. The model's strong memorization capacity often leads it to over-rely on the initial examples, limiting its adaptability to the specific case at hand. Second, the reasoning traces within ICL samples may induce cognitive dissonance with the model's inherent latent reasoning trajectories, potentially disrupting its native chain-of-thought process. Finally, given the extensive length of both target and auxiliary responses, the inclusion of few-shot exemplars introduces excessive contextual overhead, which degrades the model's ability to maintain focus on the critical task parameters.

### C.4. Additional Robustness Tables

This section reports the two robustness checks omitted from the main text for space. Table 7 changes the subset size, and Table 8 changes the judging rubric to a correctness-only prompt. Together with the auxiliary-model and resampling-policy checks in Tables 4 and 5, these results test whether the gains are sensitive to auxiliary-model choice, random subset selection, subset size, or prompt design.

---

**Algorithm 2** LLM-Based One-Step Algorithm (Zero-Shot/ICL)

---

1: **Input:** Dataset $\mathcal{D}$ with $M + 1$ auxiliary samples per instance:

$$\mathcal{D} = \left\{ x_i, y_i, g_i, \{(w_{1ij}, w_{2ij}, v_{ij})\}_{j=1}^{M+1} \right\}_{i=1}^{N}.$$

2: **Define Semantic Regressor:** Let $\tau_{\text{LLM}}(x, z)$ be the prediction score generated by a pre-trained LLM (e.g., Gemini3-flash) using a zero-shot or ICL prompt (as defined in Remark 3.5).

3: **for** $i = 1, \ldots, N$ **do**

4:     **(Integrated regression)** Approximate the integral using the last $M$ samples ($j = 2 \ldots M + 1$) via LLM inference:

$$\widehat{m}(x_i) = \frac{1}{M} \sum_{j=2}^{M+1} \tau_{\text{LLM}}(w_{1ij}, w_{2ij}, v_{ij}, x_i).$$

5:     **(Influence Score)** Compute the score using the first auxiliary sample ($j = 1$) for correction:

$$\widehat{\psi}_i = \widehat{m}(x_i) + \phi(y_i, g_i) - \tau_{\text{LLM}}(w_{1i,1}, w_{2i,1}, v_{i,1}, x_i).$$

6: **end for**

7: **Output:**

$$\widehat{\theta}^{\text{1-step}} = \frac{1}{N} \sum_{i=1}^{N} \widehat{\psi}_i.$$

---

| Model | Naive% | One-step% | Improve% |
|---|---|---|---|
| ✳ Claude-Sonnet-4.5 | 80.00 | 84.00 | +2.67 |
| ☯ DeepSeek-V3.2(Thinking) | 92.00 | 89.86 | +1.86 |
| ✦ Gemini-3-Flash-Preview | 100.00 | 98.26 | +1.74 |
| ⑤ GPT-5.2 | 100.00 | 97.59 | +2.41 |
| ✕ Grok-4-1-Fast-Reasoning | 88.00 | 85.86 | +0.53 |

*Table 7.* AIME2025 random $N = 25$ subset with GPT-5 mini and DeepSeek-V3.2.

The correctness-only rubric used for Table 8 is shown below.

---

**Correctness-Only Prompt for Auxiliary Data Evaluation**

**User Prompt:** Given a problem and two different solutions, judge them ONLY by final-answer correctness.
Problem:{question}

Solution 1:{answer1}
Solution 2:{answer2}

Rules:
- Use correctness as the only decision criterion.
- Ignore clarity, style, completeness, and elegance unless they directly affect correctness.
- If exactly one solution is correct, prefer that solution.
- If both solutions are correct or both are incorrect, this is a tie and break the tie randomly.

IMPORTANT: You must respond with ONLY a JSON object, no other text before or after. The JSON must have exactly this structure:
{
"answer1_correct": true,
"answer2_correct": false,
"tie_case": false,
"preference": "answer1",
"reasoning": "Brief explanation"
}

The "preference" field must be exactly one of:
- "answer1"
- "answer2"
- "tie"
If it is a tie, set "tie_case" to true and set "preference" to "tie".
The "reasoning" field should be a brief explanation (1-2 sentences).

JSON response:

---

| Model | Naive% | One-step% | Improve% |
|---|---|---|---|
| ✦ Gemini-3-Flash-Preview | 93.33 | 95.88 | +2.54 |
| 🐳 DeepSeek-V3.2(Thinking) | 86.67 | 90.91 | +2.42 |
| ❎ Grok-4-1-Fast-Reasoning | 80.00 | 84.69 | +4.69 |
| ✳ Claude-Sonnet-4.5 | 66.67 | 77.02 | +10.36 |
| ⑤ GPT-5.2 | 93.33 | 97.45 | +2.55 |

*Table 8.* AIME2025 robustness with a correctness-only judging rubric, using GPT-5 mini and DeepSeek-V3.2 for auxiliary generation.

## C.5. Sample Size Analysis: N=100 vs N=200 on GSM8K

In addition to the superior accuracy at fixed budgets, we also investigate the impact of sample size on the estimation accuracy by comparing the one-step estimator performance between $N = 100$ and $N = 200$ samples on the GSM8K benchmark. As illustrated in Table 9, larger sample sizes yield more precise approximations of the target metric, demonstrating the scalability and stability of the EIF framework under larger sample regimes.

**Key Observations.** Increasing sample size from $N = 100$ to $N = 200$ on the GSM8K dataset yields more stable estimates:

- **Improved Accuracy:** With $N = 200$, estimates align more closely with the ground truth (GT) in 18 out of 20 cases (90%) compared to $N = 100$. Furthermore, the average absolute error dropped from 0.96% to 0.33%, representing a significant 66% reduction.

- **Reduced Variance:** Larger sample sizes significantly enhance the stability of high-variability models, which is evident

| Model | GT% | Config 1 | | | Config 2 | | |
|---|---|---|---|---|---|---|---|
| | | N=100 | N=200 | Improv. | N=100 | N=200 | Improv. |
| ✦ Gemini-3-Flash-Preview | 97.88 | 97.90 | 97.92 | -0.02% | 98.30 | 97.84 | +0.38% |
| ⑨ GPT-5.2 | 97.50 | 97.65 | 97.60 | +0.05% | 97.00 | 97.31 | +0.31% |
| ✳ Claude-Sonnet-4.5 | 97.50 | 96.70 | 97.67 | +0.63% | 98.15 | 97.24 | +0.39% |
| ∞ Llama-3.3-70B-Instruct | 96.44 | 96.00 | 96.60 | +0.28% | 95.70 | 96.67 | +0.51% |
| 🐋 DeepSeek-R1-Distill-Llama-70B | 95.83 | 95.50 | 95.88 | +0.28% | 94.50 | 96.50 | +0.66% |
| 🜋 QwQ-32B-Preview | 95.45 | 93.85 | 95.05 | +1.20% | 94.35 | 96.14 | +0.41% |
| ✖ Grok-4-1-Fast-Reasoning | 95.07 | 92.65 | 93.95 | +1.30% | 91.20 | 95.12 | +3.82% |
| 🐋 DeepSeek-V3.2(Thinking) | 95.00 | 94.00 | 95.05 | +0.95% | 92.90 | 93.59 | +0.69% |
| 🐋 DeepSeek-R1-Distill-Qwen-32B | 93.71 | 93.90 | 93.32 | -0.20% | 93.35 | 94.09 | 0.00% |
| 🜋 Qwen3-Next-80B-A3B-Instruct | 93.48 | 93.65 | 93.45 | +0.14% | 94.30 | 92.84 | +0.18% |

*Table 9.* GSM8K: $N = 100$ vs $N = 200$ comparison. Let $\widehat{\theta}_N^{\text{1-step}}$ denote the one-step estimator with sample size $N$, and $\theta_{\text{GT}}$ the ground truth accuracy from the full dataset. Improv. $= |\widehat{\theta}_{100}^{\text{1-step}} - \theta_{\text{GT}}| - |\widehat{\theta}_{200}^{\text{1-step}} - \theta_{\text{GT}}|$ denotes the reduction in absolute error; positive values indicate that $N = 200$ yields estimates closer to $\theta_{\text{GT}}$. $N = 200$ achieves closer estimates in 18/20 cases (90%). Top 3 in GT% and $N$=200 columns: 1st , 2nd , 3rd .

| Model | GT% | Config 1 | Config 2 |
|---|---|---|---|
| ✦ Gemini-3-Flash-Preview | 90.40 | 88.90 | 89.00 |
| ⑨ GPT-5.2 | 86.36 | 83.60 | 84.10 |
| 🐋 DeepSeek-V3.2(Thinking) | 84.85 | 84.70 | 83.10 |
| ✖ Grok-4-1-Fast-Reasoning | 83.33 | 80.70 | 81.80 |
| ✳ Claude-Sonnet-4.5 | 82.83 | 84.70 | 80.70 |
| 🜋 Qwen3-Next-80B-A3B-Instruct | 76.26 | 74.90 | 75.00 |
| 🐋 DeepSeek-R1-Distill-Llama-70B | 59.09 | 58.90 | 58.60 |
| 🜋 QwQ-32B-Preview | 56.06 | 54.70 | 55.00 |
| ∞ Llama-3.3-70B-Instruct | 46.97 | 46.60 | 46.10 |
| 🐋 DeepSeek-R1-Distill-Qwen-32B | 43.43 | 42.00 | 42.00 |

*Table 10.* GPQA Diamond one-step estimator results using the full $N = 198$ dataset to demonstrate improved model ranking. GT% denotes ground truth accuracy. Config 1 uses GPT-5 mini for both auxiliary models; Config 2 uses GPT-5 mini and DeepSeek-V3.2. Top 3 accuracies are highlighted: 1st , 2nd , 3rd .

in Grok's performance: under Config 1, the error rate dropped from 2.42% to 1.12%, while in Config 2, it sharply declined from 3.87% to a mere 0.05%.

- **Consistent Across Configurations:** The two benefits identified above are robust across different choices of auxiliary models, demonstrating the generalization of sample size impact within our estimation framework.

## C.6. Model Re-ranking Results

The combination of high statistical efficiency and accurate scaling positions our one-step estimator as an ideal metric for large-scale benchmarking. As established in Section 5.2.2, while the ground truth in our experiment is defined by the naive estimator (2.1) applied to the full dataset, we can expect our EIF-based estimator to yield results that closely approximate the true model performance, providing a high-fidelity representation of the intrinsic ranking even with limited data.

From the estimation results on the whole datasets, our EIF-based estimator shows a significantly enhancement on the discriminative resolution of model performance evaluation. For example, in Table 11 for the AIME 2025 benchmark: while GPT-5.2 and Gemini-3-Flash-Preview yield identical empirical accuracies based on ground truth (both 96.67%), which actually is evaluated with pass@1 metric, our one-step estimator (N=30) reveals a performance advantage for GPT-5.2 (96.88% and 97.65% under Config 1 and 2, respectively) over Gemini-3-Flash-Preview (92.67% and 95.87% under Config 1 and 2, respectively). This suggests that the EIF method can capture more latent quality nuances and statistical confidence that traditional point estimates may overlook.

| Model | GT% | Config 1 | Config 2 |
|---|---|---|---|
| ⑤ GPT-5.2 | 96.67 | 96.88 | 97.65 |
| ✦ Gemini-3-Flash-Preview | 96.67 | 92.67 | 95.87 |
| 🐳 DeepSeek-V3.2(Thinking) | 90.00 | 86.41 | 89.22 |
| ⓧ Grok-4-1-Fast-Reasoning | 86.67 | 81.83 | 89.01 |
| ✳ Claude-Sonnet-4.5 | 83.33 | 83.00 | 85.00 |
| ◬ Qwen3-Next-80B-A3B-Instruct | 73.33 | 77.63 | 78.66 |
| 🐳 DeepSeek-R1-Distill-Llama-70B | 53.33 | 47.26 | 56.22 |
| 🐳 DeepSeek-R1-Distill-Qwen-32B | 53.33 | 51.63 | 50.83 |
| ◬ QwQ-32B-Preview | 30.00 | 24.56 | 35.24 |
| ∞ Llama-3.3-70B-Instruct | 6.67 | 7.58 | 9.40 |

*Table 11.* AIME 2025 one-step estimator results using the full $N = 30$ dataset to demonstrate improved model ranking. GT% denotes ground truth accuracy. Config 1 uses GPT-5 mini for both auxiliary models; Config 2 uses GPT-5 mini and DeepSeek-V3.2. Top 3 accuracies are highlighted: 1st , 2nd , 3rd .

## C.7. Prompt Templates and Data Samples

In this section, we detail the prompting architecture developed for constructing and assessing auxiliary information. This includes the formal templates used in our experiments, accompanied by some samples of auxiliary regressor outputs and model preference rankings.

### C.7.1. PROMPT TEMPLATES

---

**Prompt for Auxiliary Data Generation on AIME 2025**

**User Prompt:** Solve the following AIME problem. Please provide the explanation of reasoning followed by the final numerical answer.

Problem: {question}

Format your response as follows:
- Start with "Reasoning:" followed by several sentences explaining your thought process
- Then write "Final Answer:" followed by ONLY the numerical answer
- Please keep your explanation concise and to the point

Example:
Reasoning: We need to find values where the divisibility condition holds. Testing systematically, we find that b=21 and b=49 satisfy the requirement;
Final Answer: 70.

---

**Prompt for Auxiliary Data Evaluation**

**User Prompt:** Given a problem and two different solutions, determine which solution is better.
Problem:{question}

Solution 1:{answer1}
Solution 2:{answer2}

Please analyze both solutions and determine which one is better based on:
- Correctness of the final answer
- Clarity of reasoning
- Completeness of the solution

IMPORTANT: You must respond with ONLY a JSON object, no other text before or after. The JSON must have exactly this structure:
"preference": "answer1",
"reasoning": "Brief explanation"

The "preference" field must be exactly either "answer1" or "answer2" (nothing else).
The "reasoning" field should be a brief explanation (1-2 sentences).

JSON response:

**Prompt for $\tau$ Prediction (Few-Shot)**

**Few-Shot Examples:**

Example {i}:
Problem: {question}

Chain 1: {w1}

Chain 2: {w2}

Target model's judgment: {preferred_chain} is better.

Ground truth: The target model answered this problem {ground_truth}.

FINAL_PROBABILITY: {answer}

⋮                                                                                          *(repeated for k examples)*

**Test Query:**

Problem: {question}

Chain 1: {w1}

Chain 2: {w2}

Target model's judgment: {preferred_chain} is better.

Please analyze this case based on the patterns you observed in the examples above, and predict whether the target model answers correctly.

Output a calibrated probability between 0 and 1 for target-model correctness.

Your analysis and final answer should end with:

FINAL_PROBABILITY: 0.000-1.000

---

**Prompt for $\tau$ Prediction (Zero-shot)**

**User Prompt:** Problem: {question}

Chain 1: {w1}

Chain 2: {w2}

Target model's judgment: {preferred_chain} is better.

Please analyze this case and predict whether the target model answers correctly.

Output a calibrated probability between 0 and 1 for target-model correctness.

Your analysis and final answer should end with:

| FINAL_PROBABILITY: 0.000-1.000 |

---

## C.7.2. AUXILIARY DATA SAMPLES

---

**GSM8K: Auxiliary Data generated by GPT-5 mini and preference generated by Deepseek-reasoner**

**"w1"**: "Interpret \"every second glass costs only 60% of the price\" to mean each even-numbered glass (2nd, 4th, \u2026) costs 60% of \$5.\n\n1. Full price per glass = \$5.\n2. Discounted price = 60% of \$5 = 0.6 \u00d7 5 = \$3.\n3. In 16 glasses there are 8 full-price and 8 discounted glasses.\n4. Total = 8 \u00d7 \$5 + 8 \u00d7 \$3 = \$40 + \$24 = \$64.\n\nKylar needs to pay \$64."

**"w2"**: "Step 1: Full price = \$5. \nStep 2: Every second glass costs 60% of \$5 \u2192 0.6 \u00d7 5 = \$3. \nStep 3: Cost per pair = \$5 + \$3 = \$8. \nStep 4: 16 glasses = 8 pairs \u2192 8 \u00d7 \$8 = \$64.\n\nAnswer: \$64."

**"preference"**: "answer1",

**"reasoning"**: "Solution 1 is clearer and more complete as it explicitly explains the interpretation of 'every second glass' and breaks down the calculation into full-price and discounted glasses."

---

## D. Formal Definitions

**Nuisance Functions.**  In semiparametric estimation, *nuisance functions* refer to infinite-dimensional parameters that must be estimated as intermediate steps to construct an estimator for the target parameter of interest, but are not themselves the primary quantities we seek to infer. In our setting, the nuisance functions are:

- $\tau(Z, X) = \mathbb{E}[\phi(Y, G) \mid Z, X]$: the *outcome regression function*, which predicts the expected metric given the input and auxiliary information.

- $m(X) = \mathbb{E}[\phi(Y, G) \mid X]$: the *integrated regression function*, obtained by marginalizing $\tau$ over the auxiliary distribution $P(Z \mid X)$.

These functions are essential for constructing the efficient influence function and achieving optimal estimation efficiency, but estimating them is a means to an end rather than the end itself.

**Reasoning Chain.**  A *reasoning chain* (also known as chain-of-thought) refers to the sequence of intermediate reasoning steps generated by a language model before arriving at a final answer. In our framework, the auxiliary information $Z = (W_1, W_2, V)$ includes two candidate reasoning chains $W_1$ and $W_2$ produced by auxiliary LLMs, along with a preference label $V$ indicating which chain is judged to be superior. These reasoning chains serve as informative proxies that are correlated with the target model's performance, enabling more efficient estimation.

**Control Variate.**  In Monte Carlo estimation, a *control variate* is an auxiliary random variable with known expectation that is correlated with the quantity of interest. By incorporating the control variate into the estimator, one can reduce variance without introducing bias. In our setting, the auxiliary information $Z$ plays an analogous role: since the discriminative signal

from verifying reasoning chains is correlated with ground-truth correctness, the EIF-based estimator achieves substantial variance reduction compared to the naive sample mean estimator.

