# OpenReview forum: "Evaluating LLMs When They Do Not Know the Answer: Statistical Evaluation of Mathematical Reasoning via Comparative Signals"
_ICML.cc/2026/Conference — ICML 2026 regular_

### Official Review · Reviewer_jGN3 · 2026-02-26

**Soundness:** 2
**Presentation:** 3
**Significance:** 3
**Originality:** 3
**Overall Recommendation:** 4
**Confidence:** 4

**Summary:**

The paper proposes a statistically efficient way to estimate an LLM’s score on math benchmarks, where the ground truth data is scarce.  The key idea is to augment the usual correctness metric $\phi(Y,G)$ with comparative signals: for each problem $X$, generate two auxiliary solution chains $W_1, W_2$ and use the target model to produces a preference score $V$ indicating which chain is better. These auxiliary signals $Z=(W_1,W_2,V)$ are treated as control variates. The authors build the efficient influence function for $\theta=\mathbb{E}[\phi(Y,G)]$ under the specified model and derive one-step estimator that incorporates the information from $p(Z\mid X)$. They prove asymptotic normality and that the estimator achieves the semi-parametric efficiency bound; moreover, the asymptotic variance is strictly smaller than the naive sample mean whenever (Z) contains non-redundant information about correctness beyond (X).

On the empirical side, the paper reports improved estimation accuracy (relative to the full-dataset naive estimate used as “ground truth”) and more stable rankings on GPQA Diamond, AIME 2025, and GSM8K.

**Compliance With Llm Reviewing Policy:**

Affirmed.

**Final Justification:**

The authors have answered all my concerns during the rebuttal. Therefore, I am raising my score.

Note: I hope the authors integrate all the additional experiments and ablations into the final version of the manuscript.

**Key Questions For Authors:**

1. **How do you prevent cheating?** If the evaluation prompts, auxiliary models, and sampling is handled by developers, this could lead to a wide oscillations in the reported results. It seems it's easy to game the metric by trying different auxiliary models, resampling, and selection bias.

2. **How stable is the “known $p(Z\mid X)$” assumption operationally?** If a provider updates a model or changes hidden prompts, do you treat that as a new $p(Z\mid X)$? Any proposed protocol to detect/handle drift?

3. **Budget trade-off:** For a fixed number of target-model tokens/calls, how does your estimator compare to simply sampling multiple $Y$’s per $X$ (e.g., estimating pass@k or mean accuracy under repeated decoding)? A plot of variance vs. total tokens would be useful.

4. **Dependence on auxiliary-chain quality:** How sensitive are results to (i) very weak auxiliary chains, (ii) very strong auxiliary chains that often contain the correct answer, or (iii) adversarially misleading chains? Do you have diagnostics that predict when $Z$ is informative enough (e.g., estimating $\mathrm{Var}(\hat\tau(X,Z))$)?

5. **Target-as-judge bias:** Given your preference rubric includes clarity/completeness, did you test variants that focus only on final-answer correctness? How often does the target model prefer a more fluent but incorrect chain?

**Limitations:**

yes

**Strengths And Weaknesses:**

## Strengths

* **Interesting framing.** The authors use the influence function framework to incorporate additional information into the benchmark scoring rule. It seems to be the right abstraction for “small N + stochastic outputs,” and it makes the goal (variance reduction + uncertainty quantification) explicit. The EIF derivation and one-step construction connect directly to semi-parametric efficiency, and the paper states clear conditions for asymptotic normality and strict efficiency gain over naive averaging.

* **Practical algorithm.** The authors provide a practical implementation, which relies on the cross-fitting. The cross-fitted estimator (Algorithm 1) is concrete and matches standard debiased/DML practice: **1)** estimate $\hat\tau = E[\phi(Y, G) | X, Z]$, which measures how the correctness metric depends on the auxiliary signal **2)** integrate out the auxiliary variables via Monte Carlo $m(x)$ **3)** compute influence score as $\psi = \hat\tau - \hat m + \phi$.

* **Empirical coverage across multiple benchmarks + configurations.** The experiments span very small (AIME 2025) to larger (GSM8K) and evaluate both homogeneous and heterogeneous auxiliary sources; the appendix also includes sample-size scaling and re-ranking analyses.

* **Good operational detail in the appendix.** Prompt templates for generating auxiliary chains, collecting preferences, and producing (\tau) predictions are included, which helps reproducibility.

---

## Weaknesses

* **Estimating scores via repetitive prompting of an LLM as a judge could open the doors to cherry-picking and benchmark gaming.** The final score on the benchmark is obtained via the Monte Carlo estimate term $m = \frac{1}{M} \sum \hat \tau (Z, X)$. If developers/practitioners have access to the auxiliary samples $Z$, they can superficially select samples that inflate $m$, e.g., doing something like $(Z_i)_{i\in I} = \max_I \tau(Z_i, X) $.

* **The “known (p(Z\mid X))” assumption is fragile in real deployments.** The theory leans on treating the auxiliary-generation mechanism as known and fixed, justified by “we control prompting and can resample.”  In practice, API/model updates, hidden system prompts, sampling nondeterminism differences across providers, and throttling can make the effective (p(Z\mid X)) non-stationary or partially unobservable. Even if you can Monte Carlo sample, it may not correspond to a stable target distribution over time.

* **Bias/robustness hinges on nuisance quality, but the “semantic regressor” is not analyzed as an estimator.** In the small-sample pathway, (\tau_{\text{LLM}}) is treated as a plug-in proxy.  If (\tau_{\text{LLM}}(x,z)) is systematically miscalibrated, the one-step correction can become biased (and not just higher-variance). The paper emphasizes asymptotics under (\hat\tau \to \tau), but does not tightly connect that to the behavior of an LLM-based scorer.

* **Compute / query budget accounting is underdeveloped.** The method introduces multiple additional calls per instance: generating (W_1,W_2), eliciting (V), and computing (\tau) repeatedly for Monte Carlo integration (they use (M=10)).  It’s not obvious when this is cheaper than alternatives like repeated sampling of (Y) (self-consistency) or collecting more labeled items (when possible). A cost–variance trade-off curve would matter for adoption.

* **Using the target model as judge can introduce systematic style preferences.** The preference prompt explicitly scores “correctness, clarity, completeness,” and forces a choice.  For some models, “clarity” or verbosity may correlate with preference independent of correctness, potentially creating spurious control-variate signal and, worse, bias in finite samples.

* **Failure cases are not deeply diagnosed.** The tables show instances where the one-step estimate is worse than naive for some models/configurations (negative “Improv.” values appear), yet the paper mostly emphasizes consistent gains.  A systematic analysis of when/why auxiliary signals hurt (e.g., low judge reliability, adversarial auxiliary chains, distribution shift across problem types) would strengthen credibility.

* **“Ground truth” is approximated by full-dataset naive accuracy, which is still noisy on small datasets.** Defining GT as the naive estimator on the full benchmark is pragmatic, but on AIME (30 problems) that baseline itself has substantial uncertainty; improvements measured against it conflate estimator improvement with GT noise.

---

> ### Author Rebuttal · Authors · 2026-03-31
>
> Dear Reviewer jGN3,
> Thank you for your comments. We would like to address each concern below.
>
> ## Q1&W1
> We would like to note that this concern applies broadly to all evaluation pipelines rather than being specific to our method. Our contribution is that under a fixed evaluation protocol, incorporating auxiliary signals yields strictly lower-variance estimates than standard approaches. In our experiments, we fix the evaluation prompts, auxiliary generation procedure, and all sampling details across methods to ensure a fair comparison and observe consistent improvements over the baseline. All implementation details, including prompts and how auxiliary signals are generated, are fully specified in the Appendix for reproducibility.
>
> ## Q2&W2
> We clarify that “$p(Z|X)$ is known” is not an artificial modelling assumption, but a direct consequence of how LLM evaluation is conducted. The auxiliary signals $Z$ are generated by querying models under a specified prompt and decoding configuration; therefore, the conditional distribution $p(Z|X)$ is fully determined by this generation procedure.
>
> Operationally, for any fixed evaluation run, i.e., with a fixed model, prompt, and decoding settings, $p(Z|X)$ is stable and well-defined. This remains true regardless of hidden system prompts, as long as the underlying generation pipeline is unchanged during the run. In this sense, “known $p(Z|X)$” reflects that the evaluator has full control over (and can repeatedly sample from) the data-generating mechanism.
>
> ## Q3&W4
> We first clarify that pass@k and mean accuracy correspond to different estimands. Our goal is not to introduce a new metric, but to provide a more statistically efficient estimator for a fixed target metric (e.g., accuracy). Therefore, our method is complementary to, rather than a substitute for, pass@k.
>
> To compare with repeated decoding, we conduct an equal-budget experiment on GPQA Diamond. Our method uses 1 target response and 11 comparisons, while the repeated-Y baseline uses 12 target responses. Our method's results(C1, C2) can be seen at line 387 in the paper; the table below reports the Repeated-Y and the gap to the ground truth.
> | Model | 12-Y (Gap to GT) |
> |:-:|:-:|
> | gemini3flash | 88.50 (1.90) |
> | gpt5.2 | 85.33 (1.03) |
> | deepseekV3.2 | 83.17 (1.68) |
> | grok4.1fast | 79.50 (3.83) |
> | claude sonnet4.5 | 76.83 (6.00) |
>
> Under the same budget, our method is closer to the ground truth than repeated-Y in 4 out of 5 models (for both C1 and C2), demonstrating better sample efficiency.
> More broadly, the total uncertainty in LLMs evaluation arises from two sources: (i) stochasticity of model outputs, and (ii) finite benchmark size. Repeated decoding primarily reduces (i), and with sufficient budget can drive it to near zero. As a core contribution, our method targets (ii) by extracting additional information per sample via auxiliary signals. Even when decoding noise is negligible, our estimator achieves strictly lower variance with respect to the data distribution, and is therefore complementary to repeated sampling.
>
> ## Q4
> Our current experiments cover two non-adversarial regimes (C1 and C2), where auxiliary signals have different strengths but are both informative, and we consistently observe improved estimation accuracy.
>
> The behavior of our estimator aligns with the theory. When $Z$ is informative ($\tau(X,Z)$ varies with $Z$), we obtain variance reduction. In the extreme case where $Z$ is adversarial and thus completely uninformative (i.e., $\tau(X,Z)=m(X)$), the estimator reduces to the naive baseline, and no harm is incurred.
>
> ## Q5&W5
> The main contribution of our paper is to show that auxiliary information can improve evaluation, and our experiments already show that the proposed estimator improves over the baseline. Alternative rubrics may further help our estimator, but is not the focus of this work. We will explore alternative rubrics and auxiliary signals as an important direction for future work.
>
> ## W3
> Theoretically, our method requires this nuisance estimator to be reasonably accurate.
> Empirically, our results show that even when $\hat\tau$ is implemented via an off-the-shelf LLM, the proposed estimator consistently improves over the baseline. This indicates that the method is robust to moderate misspecification and can deliver practical gains without requiring a perfectly calibrated nuisance model.
>
> ## W6
> Based on our theory, the estimator can’t perform better in every single instance. A smaller variance does not mean the error will be smaller every time; rather, it means that the error will be smaller on average. Therefore, it is expected and normal for the estimator to underperform in a few occasional cases.
>
> ## W7
> We clarify that “ground truth” refers to the accuracy computed on the full dataset, while the reported estimators are computed on a smaller subset (e.g., 15 problems). This setup is intended to simulate the scenario where the full dataset serves as the population quantity.

---

> > ### Author Rebuttal · Reviewer_jGN3 · 2026-04-01
> >
> > Thank you for the rebuttal. The additional details address some of my questions. However, some of my main concerns still remain:
> > - On gaming / cherry-picking: saying that “all evaluation pipelines can be gamed” does not address the specific vulnerability introduced here. My concern is precisely that the reported score now depends on choices of auxiliary models, prompts, resampling policy, and selection protocol. A convincing response would require a stricter evaluator-side protocol or evidence that results are robust to these choices.
> > - The response on auxiliary quality is too idealized. The claim that adversarial or uninformative $Z$ “causes no harm” is a population-level statement under the model, but in finite samples the method can underperform, and the paper itself shows negative improvement entries for some model/configuration pairs.
> > - Finally, my concern about target-as-judge bias is not resolved by saying alternative rubrics are future work. The current evaluation prompt explicitly asks the judge to compare solutions using not only correctness, but also clarity and completeness, which leaves open the possibility that stylistic preferences are being used as control-variate signal. I think a more careful ablation on the judge prompt would be valuable.

---

> > > ### Author Response · Authors · 2026-04-02
> > >
> > > Dear Reviewer jGN3,
> > >
> > > Thank you for your follow-up comments. We clarify them more precisely below.
> > >
> > > First, we would like to clarify a potential misunderstanding. Our contribution is not that arbitrary auxiliary signals $Z$ universally improve evaluation. Rather, our paper establishes:
> > > 1. A principled estimator that incorporates a given auxiliary signal $Z$ via a semiparametric control variate construction, yielding efficiency gains when $Z$ is informative.
> > > 2. A concrete instantiation of $Z$ via comparative signals motivated by the generation–verification gap.
> > >
> > > That said, we appreciate the reviewer’s suggestion, and we have conducted additional experiments to assess robustness across different choices of $Z$. In the following, we answer your detailed questions one by one below
> > >
> > > ## gaming / cherry-picking
> > >
> > > Regarding the evaluator-side choices, we explicitly vary all four components mentioned by the reviewer:
> > > - Auxiliary models: we add a new setting where $W_1$ and $W_2$ are generated by GPT5mini and QwQ-32B, respectively. The results are summarized in Table 1.
> > > - Prompts: we add a new correctness-only rubric (see more details in Point 3 and Table 4)
> > > - Resampling policy: we repeat the analysis on a resampled subset of 15 problems drawn from the full AIME 2025 dataset using a different random seed. The results are summarized in Table 2.
> > > - Selection protocol: we vary the subset size by considering a larger subset of 25 problems drawn from the full AIME dataset to examine the method’s stability with respect to subset size. See Table 3.
> > >
> > > Table 1. AIME GPT5mini+QwQ-32B-Preview
> > >
> > > | Model | Naive | Proposed Method | Improve |
> > > |:-:|:-:|:-:|:-:|
> > > | gemini3flash | 93.33 | 96.68| +3.32 |
> > > | deepseekV3.2 | 86.67 | 87.72| +1.05 |
> > > | grok4.1fast | 80.00 | 86.21| +6.22 |
> > > | gpt5.2 | 93.33 | 97.51| +2.48 |
> > > | claude-sonnet4.5 | 66.67 | 72.95 | +6.29 |
> > >
> > > Table 2. AIME Different seed with GPT5mini+DeepseekV3.2
> > >
> > > | Model | Naive | Proposed Method | Improve |
> > > |:-:|:-:|:-:|:-:|
> > > | claude-sonnet4.5 | 80.00 | 82.67 | +2.66 |
> > > | deepseekV3.2 | 86.67 | 91.33 | +2.00 |
> > > | gemini3flash | 100.00 | 98.00 | +2.00 |
> > > | gpt5.2 | 93.33 | 98.67 | +1.33 |
> > > | grok4.1fast | 93.33 | 84.00 | +4.00 |
> > >
> > > Table 3. AIME Random $N=25$ subset with GPT5mini+DeepseekV3.2
> > >
> > > | Model | Naive ($N=25$) | Proposed Method ($N=25$) | Improve |
> > > |:-:|:-:|:-:|:-:|
> > > | claude-sonnet4.5 | 80.00 | 84.00 | +2.67 |
> > > | deepseekV3.2 | 92.00 | 89.86 | +1.86 |
> > > | gemini3flash | 100.00 | 98.26 | +1.74 |
> > > | gpt5.2 | 100.00 | 97.59 | +2.41 |
> > > | grok4.1fast | 88.00 | 85.86 | +0.53 |
> > >
> > > Across Tables 1–3, we observe that: The proposed estimator consistently remains closer to the ground truth than the naive estimator across configurations. In addition, improvements are stable across heterogeneous auxiliary pipelines, not tied to a specific choice.
> > >
> > > This empirically supports that the gains are not driven by cherry-picked configurations, but persist under perturbations of the evaluation protocol. We will also release code to ensure full transparency and reproducibility of all protocol choices.
> > >
> > > ## auxiliary quality
> > > Regarding auxiliary quality, we agree that finite-sample degradation can occur when $Z$ is weak or adversarial. This is consistent with our theory: the efficiency gain depends on the informativeness of $Z$, and in the worst case the estimator may not improve over naive in finite samples. We will clarify this point explicitly in the revision. More generally, our results are variance-reduction guarantees, not pointwise dominance guarantees. A lower variance does not imply uniformly smaller error for every realization; rather, it ensures improved performance on average. Consequently, it is expected that the estimator may underperform in a small number of instances, even when it achieves strictly lower variance overall.  We will also clarify this in the revision.
> > >
> > > ## target-as-judge bias
> > >
> > > We added a judging rubric that only focuses on correctness and removes clarity and completeness, from the prompt:
> > >
> > > Prompt:
> > > Rules:
> > > - Use correctness as the only decision criterion.
> > > - Ignore clarity, style, completeness, and elegance unless they directly affect correctness.
> > > - If exactly one solution is correct, prefer that solution.
> > > - If both solutions are correct or both are incorrect, this is a tie and break the tie randomly.
> > >
> > > Table 4. AIME Correctness-only with GPT5mini+DeepseekV3.2
> > >
> > > | Model | Naive | Proposed Method | Improve |
> > > |:-:|:-:|:-:|:-:|
> > > | gemini3flash | 93.33 | 95.88 | +2.54 |
> > > | deepseekV3.2 | 86.67 | 90.91 | +2.42 |
> > > | grok4.1fast | 80.00 | 84.69 | +4.69 |
> > > | claude-sonnet4.5 | 66.67 | 77.02 | +10.36 |
> > > | gpt5.2 | 93.33 | 97.45 | +2.55 |
> > >
> > > This result suggests that the correctness-only rubric is also informative and makes our proposed method outperform the baseline. Thanks to your comment, we will also add these results in the revision to further demonstrate the gain of our method under different judging rubrics.

---

### Official Review · Reviewer_khdH · 2026-03-11

**Soundness:** 4
**Presentation:** 3
**Significance:** 3
**Originality:** 4
**Overall Recommendation:** 5
**Confidence:** 2

**Summary:**

This paper proposes a new method for evaluating mathematical reasoning in LLMs. Compared to the *naive estimator*, which averages the accuracy metric over instances of pairs prompt -> response vs ground truth, the new evaluation method adds information from multiple pairwise comparisons procedure, where the target model is prompted to compare two proposed solutions by auxiliary models. This procedure yields strictly lower variance of the estimator, thus improving performance of LLM benchmarks on limited data.

**Compliance With Llm Reviewing Policy:**

Affirmed.

**Key Questions For Authors:**

None.

**Limitations:**

Yes.

**Strengths And Weaknesses:**

The novel method is well justified, described, and evaluated.

The method's target is to reduce variance of the accuracy estimates obtained from limited benchmarks datasets. It does so correctly and efficiently.

A more broader question regarding ranking LLMs based on accuracy metrics should be focused on the content validity of these benchmarks. The current article does not address these concerns.

---

> ### Author Rebuttal · Authors · 2026-03-31
>
> Dear Reviewer khdH,
>
> Thank you for your positive feedback and for raising this important perspective. We fully agree that the content validity of current benchmarks is a fundamental issue in LLM evaluation. Our work focuses on improving the statistical efficiency of estimation given a fixed dataset; however, if a benchmark does not faithfully measure true reasoning ability, even a perfectly precise estimator cannot address that limitation.
>
> Thanks to your comment, we will add a discussion in the Limitations and Future Work section to explicitly highlight benchmark validity as an important open problem, and clarify that our framework is complementary to, rather than a replacement for, careful benchmark design.

---

> > ### Author Rebuttal · Reviewer_khdH · 2026-04-01
> >
> > I did not have many questions and there was not much to resolve to begin with.

---

> > > ### Author Response · Authors · 2026-04-02
> > >
> > > Dear Reviewer khdH,
> > >
> > > Thank you for your feedback. It is encouraging to hear that our response met your expectations. The insights regarding benchmark limitations will be a valuable addition to our revision. We appreciate your support.

---

### Official Review · Reviewer_1sUu · 2026-03-12

**Soundness:** 3
**Presentation:** 1
**Significance:** 3
**Originality:** 2
**Overall Recommendation:** 3
**Confidence:** 4

**Summary:**

In evaluating mathematical reasoning, the metric $\theta \coloneqq \mathbb{E}[\phi(Y, G)]$ may suffer from high variance when the sample size is small. Here, $Y$ denotes the answer generated by LLMs, whereas $G$ refers to the ground truth. By incorporating the statistics of auxiliary pairwise comparison of LLM-generated answers, this paper claims that $\theta$ can be orthogonally decomposed as presented in Proposition 3.3. Such a decomposition serves as the foundation for the design of their algorithms as well as the claimed variance reduction in estimating $\theta$ in Corollary 4.7.

**Compliance With Llm Reviewing Policy:**

Affirmed.

**Final Justification:**

Since most of my concerns are addressed, I raise my score accordingly. Personally, I find the theoretical presentation of this submission somewhat misleading and disrupted. I spent a lot of time going back and forth through the submission to understand what the authors intended to convey, which I consider a negative signal. If the presentation is smooth, I would have given at least 4 instead.

For example, in Corollary 4.7 which proves $\mathrm{Var}(\psi\_{\mathrm{np}}) \geq \mathrm{Var}(\psi)$, I think the whole theoretical presentation is unnecessarily complicated. By definition, $$\psi\_{\mathrm{np}} = (\psi\_{\mathrm{np}} - \mathbb{E}[\psi\_{\mathrm{np}}\mid X,Z]) + ( \mathbb{E}[\psi\_{\mathrm{np}}\mid X,Z] -  \mathbb{E}[\psi\_{\mathrm{np}}\mid X]) + \mathbb{E}[\psi\_{\mathrm{np}}\mid X]$$ (these three parts are mutually orthogonal), and $$\psi =  (\psi\_{\mathrm{np}} - \mathbb{E}[\psi\_{\mathrm{np}}\mid X,Z]) + \mathbb{E}[\psi\_{\mathrm{np}}\mid X].$$ The mutual orthogonality implies
$$\mathrm{Var}(\psi\_{\mathrm{np}}) = \mathrm{Var}(\psi\_{\mathrm{np}} - \mathbb{E}[\psi\_{\mathrm{np}}\mid X,Z]) + \mathrm{Var}( \mathbb{E}[\psi\_{\mathrm{np}}\mid X,Z] -  \mathbb{E}[\psi\_{\mathrm{np}}\mid X]) + \mathrm{Var}(\mathbb{E}[\psi\_{\mathrm{np}}\mid  X])$$
and
$$\mathrm{Var}(\psi) = \mathrm{Var}(\psi\_{\mathrm{np}} - \mathbb{E}[\psi\_{\mathrm{np}}\mid X,Z]) + \mathrm{Var}(\mathbb{E}[\psi\_{\mathrm{np}}\mid X]),
$$
where we can see that $\mathrm{Var}(\psi\_{\mathrm{np}}) >  \mathrm{Var}(\psi)$ if $\mathrm{Var}( \mathbb{E}[\psi\_{\mathrm{np}}\mid X,Z] -  \mathbb{E}[\psi\_{\mathrm{np}}\mid X]) \neq 0$, which is exactly the one used in line 802 for proving Corollary 4.7. Such a geometry is very clear in this way, and the only assumption is $\mathbb{E}|\psi\_{\mathrm{np}}|^2 < \infty$. In other words, the idea of variance reduction is nearly trivial rather than something sophisticated to demonstrate. By contrast, the presentation of this submission makes it overly complicated. I spent substantial time carefully reading the manuscript, only to realize that the underlying idea is quite simple.

**Key Questions For Authors:**

See my concerns above.

**Strengths And Weaknesses:**

**Weaknesses**
- This paper is poorly written from a theoretical perspective. When reviewing Proposition 3.3, which is key to the design of the algorithms and the overall theoretical development, I had to repeatedly move between the main text and the appendices to fully understand what it means and how it relates to the proposed algorithms. After carefully examining the result, I found that Proposition 3.3 appears to be fundamentally flawed, which in turn renders Corollary 4.7 invalid.

  - The key idea underlying the entire theoretical development is quite simple: every zero-mean random variable $X$ (including the $\psi$ in Proposition 3.3) admits a unique orthogonal decomposition

    $$
    X = (X-\mathbb{E}[X\mid Y]) + \mathbb{E}[X\mid Y]\ \text{ in the sense that }\ \langle X-\mathbb{E }[X\mid Y], \mathbb{E}[X\mid Y] \rangle \coloneqq \mathbb{E}[( X-\mathbb{E }[X\mid Y])\mathbb{E}[X\mid Y]] = 0.
    $$

    For those who may be unfamiliar with this fact,  note that
    $$
    \mathbb{E}[\mathbb{E }[X\mid Y]\mathbb{E }[X\mid Y]] = \mathbb{E}[\mathbb{E}[X\mathbb{E}[X\mid Y]\mid Y]] = \mathbb{E}[X\mathbb{E}[X\mid Y]].
    $$ Here, $\mathbb{E}[X\mid Y]$ is formally defined as $\mathbb{E}[X\mid \sigma(Y)]$, which can be constructed via the Radon-Nikodym theorem. This observation alone is sufficient to see why Proposition 3.3 is flawed and why Corollary 4.7 is invalid.

  - In line 618, the paper uses $\mathcal{T}$ to denote the space of score functions. However, this symbol has already been used in line 109 of the right column to denote the space of texts, which may cause confusion. To avoid this ambiguity, I will use
    $$
    \mathcal{S} \coloneqq \\{ \psi(Y,G) \mid \psi \in L_2(P) \text{ and } \mathbb{E}[\psi(Y,G)] = 0 \\}
    $$ to refer to the space of zero-mean score functions instead. According to the orthogonal decomposition, for every $\psi \in \mathcal{S}$,

    $$
    \psi(Y,G) = (\psi(Y,G) - \mathbb{E}[\psi(Y,G)\mid X, Z]) +  \mathbb{E}[\psi(Y,G) \mid X, Z].
    $$
    In particular, in lines 639 and 651,
    $$
    \tau(X,Z) \coloneqq \mathbb{E}[\psi(Y,G)\mid X, Z]\ \text{ and }\ m(X) \coloneqq \mathbb{E}[\psi(Y,G) \mid X].
    $$ Specifically, Proposition 3.3 claims that
    $$
    (\psi(Y,G) - \tau(X,Z)) \perp m(X).
    $$ Since such an orthogonal decomposition is unique, there must have $\tau(X, Z) = m(X)$. According to lines 611, 630, and 658, $\psi(X, Y, G, Z) = \phi(Y,G) - \theta$, the definition of which is missing in the main paper. **Substituting this definition into Proposition 3.3 immediately yields $\tau(X, Z) = m(X)$, indicating that Proposition 3.3 holds if and only if $\tau(X, Z) = m(X)$.**

  - Regarding $\tau(X, Z) = m(X)$, it does not appear to hold under Assumption 3.1. But it holds if it is $p(y,g,z,x) = p(y,g\mid x)\cdot p(z\mid x) \cdot p(x)$ instead, which I think aligns more with their introduced algorithms.

  - Corollary 3.7 states that the variance is reduced if $\tau(X, Z) \neq m(X)$, which immediately makes the orthogonal decomposition in Proposition 3.3 impossible.

- On the other hand, the theoretical perspective is not insightful as $Z$ could be any random texts, which does not explain why $Z$ should be some auxiliary pairwise comparison of LLM-generated answers.

- The Assumption 4.2 does not hold for their Algorithm 2. By definition, $\tau(X,Z) \in [0, 1]$; according to the prompt in line 1239, $\hat{\tau} \in \\{0, 1\\}$.

Overall, the proposed method is not theoretically sound to support their claimed contributions.

---

> ### Author Rebuttal · Authors · 2026-03-25
>
> Dear Reviewer 1sUu,
>
> Thank you for your detailed comments. We would like to clarify that our proof of Proposition 3.3 is indeed correct. We believe your main concern arises from a misunderstanding of the role of the efficient influence function (EIF) under a constrained semiparametric model, and we address this below.
>
> ### 1. $\psi$ vs. $\psi_{\text{np}}$
>
> You wrote:
> > "According to lines 611, 630, and 658, $\psi(X,Y,G,Z)=\phi(Y,G)-\theta$"
>
> We would like to clarify that according to the proof from lines 611-658, we have $\psi_{\text{np}}(X,Y,G,Z)=\phi(Y,G)-\theta$ (see line 630), not "$\psi(X,Y,G,Z)=\phi(Y,G)-\theta$". $\psi(X,Y,G,Z)$ is explicitly defined as the notation for the EIF (Line 166, left column). In contrast, $\psi_{\text{np}}(X,Y,G,Z)$, with the subscript $\text{np}$, is the notation for the EIF under the full nonparametric model. Introduced in Line 630, it has the form $\psi_{\text{np}}(X,Y,G,Z)=\phi(Y,G)-\theta$, which serves as a necessary technical stepping stone to derive the EIF $\psi(X,Y,G,Z)$ through projection. The subscript is explicitly used to distinguish it from our proposed $\psi$, as they are defined over different model spaces. Conflating these two distinct definitions leads to your incorrect conclusion that $\tau=m$.
>
> ### 2. The tangent space and the projection
>
> You wrote:
> > "The key idea underlying the entire theoretical development is quite simple: every zero-mean random variable $X$ (including the $\psi$ in Proposition 3.3) admits a unique orthogonal decomposition $X =(X-\mathbb{E}[X|Y])+\mathbb{E}[X|Y]$"
>
> This misunderstands our proof as simply subtracting and adding "$\mathbb{E}[X|Y]$". We clarify as follows.
>
> According to standard semiparametric inference (see, e.g., Van der Vaart, Asymptotic Statistics, 1998, Chapter 25), the most efficient estimator is constructed based on its EIF constrained to the corresponding tangent space. In our setting of LLMs evaluation, a technical challenge is that the tangent space is quite different from the classical case, as the distribution $p(z|x)$ is known (Assumption 3.1, line 140, right column). In particular, the tangent space $\mathcal{S}$ considered in the LLMs evaluation setting is the closure of all score functions associated with regular parametric submodels satisfying Assumption 3.1 (Line 140, right column). Therefore, to compute the EIF constrained on this particular tangent space $\mathcal S$, our technical novelty is to first construct two orthogonal subspaces $\mathcal T_1$ and $\mathcal T_2$, such that $\mathcal S=\mathcal T_1 \oplus \mathcal{T}_2$ and the EIF constrained on $\mathcal{T}_1$ and $\mathcal T_2$ are easier to calculate, and then add them up.
>
> To obtain $\psi$, we project $\phi(Y,G)-\theta$ (equivalently $\psi_{\text{np}}$) onto $\mathcal{S}$ to get $\psi=\mathbb{E}[\phi(Y,G)-\theta|\mathcal{S}]$, computed via the orthogonal decomposition $\psi = \mathbb{E}[\psi|\mathcal{T}_1]+\mathbb{E}[\psi|\mathcal{T}_2]$, where $\mathbb{E}[\psi|\mathcal{T}_1] = m(X) - \theta$ and $\mathbb{E}[\psi | \mathcal{T}_2]  = \phi(Y,G) - \tau(X,Z)$.
>
> Thus $\psi$ is precisely the projection of $\psi_{\text{np}}$ onto $\mathcal{S}$, rather than a mere algebraic decomposition, and it is this projection that yields the variance reduction we demonstrate.
>
> Your comment "$S= \lbrace\psi(Y,G) |\psi \in L_2(P)~\mathbb{E}[\psi(Y,G)]=0 \rbrace$" is incorrect either. $\mathcal{S}$ considered here isn’t the whole $L_2$ space generated only by $Y,G$. It is the closure of all score functions associated with regular parametric submodels satisfying Assumption 3.1.
>
> ### 3. Definitions of $\tau$ and $m$
>
> In your comment, the definitions $\tau,m$ appear to confuse $\phi$ and $\psi$. $\phi$ is the metric function. We treat this as a typo and address your concern assuming it is $\phi$.
>
> ### 4. The choice of $Z$
>
> With respect to $Z$, it cannot be just “any random text”; it must be informative, i.e., $Z$ should not be independent of $Y$ given $X$ (Line 227, right column). Our choice of pairwise comparison signals is motivated by the generation–verification gap and is supported empirically: across GPQA, AIME25, and GSM8K, incorporating these signals consistently improves evaluation accuracy over the baseline.
>
> ### 5. Output of $\tau$
>
> Regarding the output of $\tau$ in our implementation, it provides a predicted probability (a soft score in $[0,1]$), consistent with the theory.
>
> ---
>
> In summary, the key point is that our EIF is obtained via projection onto a constrained tangent space, not via a generic $L_2$ decomposition. This distinction leads to the non-degenerate structure $\tau(X,Z) \neq m(X)$ and underlies the variance reduction guarantees. We hope this clarification resolves your concern regarding our technical proof.
>
> Again, we thank you for your comments. We will add a high-level proof sketch in our proof to make it clearer, and also revise the notation to avoid overloading symbols (replacing $\mathcal{T}$ by $\mathcal{S}$) to improve clarity.

---

> > ### Author Rebuttal · Reviewer_1sUu · 2026-04-03
> >
> > - According to the authors, $\psi(X, Y, G, Z)$ is introduced as a definition rather than a derived result, and this should be stated clearly in the main paper. Under this clarification, it follows that $m(X) - \tau(X,Z) + \phi(Y,G) \neq \phi(Y,G)$. Then, it is better to point out that $\mathbb{E}[m(X) - \tau(X,Z)] = 0$ so that the metric $m(X) - \tau(X,Z) + \phi(Y,G)$ is a reasonable substitute to $\phi(Y,G)$. Moreover, I think $\psi = \mathbb{E}[\phi(Y,G) - \theta\mid \mathcal{S}]$ should be included to avoid confusion.
> >
> > -  I am fine with $Z$ being generated empirically and heuristically, although the theory does not apply solely to pairwise comparison.
> >
> > - In your experiments, does $Y$ depend on $(X,Z)$ or $X$? I went through the paper several times, perhaps it is somewhere, but I did not find it. If $Y$ depends on $X$, I do not see $Z$ is not independent of $Y$ given $X$. I may change my opinion on this part later, as I am thinking whether the internal state of LLMs would make $Y$ and $Z$ not independent given $X$ even if $Y$ depends on $X$.
> >
> > - For $\tau_{\mathrm{LLM}}$, isn't that the prompt in line 1265 only asks LLMs to output whether the target model answers correctly, which corresponds to $\\{0, 1\\}$? Or the prompt is not the one used in your experiments?

---

> > > ### Author Response · Authors · 2026-04-03
> > >
> > > Thank you for your follow-up comments. We address each point below.
> > >
> > > 1.We would like to clarify that the notation $\psi$ is defined as the notation for the EIF, and the expression $\psi=(m(X)-\theta) - (\tau(X,Z) - \phi(Y,G))$ is a derived result established in Proposition 3.3, not a definition. We will make this distinction explicit in the main text.
> > >
> > > Thanks to your comment, we will point out $\mathbb{E}[m(X) - \tau(X,Z)]=0$ after we introduce $\tau(X,Z)$ and $m(X)$ in equations (3.2) and (3.3).
> > >
> > > In addition, in the proof of Proposition 3.3, we will expand the sentence “The EIF in our constrained model is the projection of $\psi_{np}$ ” by adding the formula $\psi=\mathbb{E}[\psi_{np}|\mathcal{S}]=\mathbb{E}[\phi(Y,G)-\theta|\mathcal{S}]$ to improve clarity.
> > >
> > > 2.Thank you, we are glad our previous response addressed your concern.
> > >
> > > 3.In the experiment, the answer $Y$ is generated by inputting the target model with the question $X$. We will include the detailed prompt in the revision. The auxiliary signal $Z$ consists of $(V, W_1, W_2)$, where $V$ is produced by the same target model through a comparison of the generated reasoning answers $W_1$ and $W_2$.
> > >
> > > Consequently, $Y$ and $Z$ are generally dependent conditional on $X$. Intuitively, both reflect the latent capabilities (e.g., reasoning ability) of the same underlying model given $X$, which induces dependence.
> > >
> > > 4.Thank you for catching this. The prompt referenced in the earlier version is outdated. In our current experiments, the judge outputs a calibrated probability in $[0,1]$ for target-model correctness using the prompt: “Output a calibrated probability between 0 and 1 for target-model correctness.” We will update the prompt description in the paper and release the full code to ensure transparency and reproducibility.
> > >
> > > We would appreciate it if you could update your score if your concerns are addressed, and please let us know if you have any further comments.

---

### Official Review · Reviewer_AT79 · 2026-03-12

**Soundness:** 3
**Presentation:** 3
**Significance:** 4
**Originality:** 2
**Overall Recommendation:** 5
**Confidence:** 3

**Summary:**

This paper tackles the problem of evaluating LLMs on smaller benchmark datasets which are hard, that lead to noisy and high variance accuracy estimates. The primary method is to additionally have auxiliary models also output answers to questions, and have the evaluated model also judge between these auxiliary answers. An elegant statistical framework is formulated to combine the original prediction with these auxiliary judgements to develop an estimator the achieves a semi-parametric bound and reduces variance over standard sample averaging. Thorough numerical experiments corroborate this framework and theory.

**Compliance With Llm Reviewing Policy:**

Affirmed.

**Final Justification:**

I had no major concerns, and the authors' rebuttal appropriately addressed my concerns, so I maintain my score of 5.

**Key Questions For Authors:**

1. What about other baselines beyond naive sample average, like Bayesian estimators or other variance reduction techniques?

**Limitations:**

yes

**Strengths And Weaknesses:**

Strengths:
1. The original problem is very well motivated and is of high significance in the current research landscape.
2. The proposed solution is grounded on an elegant and well structured statistical framing.
3. The resulting practical implementation of the method is also fairly well explained, and the numerical results are strong.

Weaknesses:
1. There is some gap between theory and practice particularly on the estimation of the nuisance $\tau$. As per my understanding, the theory relies on good estimate, but the practical implementations uses a zero shot prompt of an LLM, and it is unclear how good this is.

---

> ### Author Rebuttal · Authors · 2026-03-31
>
> Dear Reviewer AT79,
>
> Thank you for your positive feedback and for recognizing both the importance of the problem and the statistical structure of our framework.  We address your insightful questions regarding the estimation of the nuisance parameter and the choice of baselines below.
>
> We agree that the quality of the nuisance estimator $\hat\tau$ is important. Our theoretical guarantees require $\hat\tau$ to be a reasonably accurate approximation of $\tau$, and the asymptotic results are stated under this condition.
> Empirically, our results show that even when $\hat\tau$ is implemented via an off-the-shelf LLM, the proposed estimator consistently improves over the baseline. This indicates that the method is robust to moderate misspecification and can deliver practical gains without requiring a perfectly calibrated nuisance model.
>
> Regarding alternative baselines such as Bayesian estimators or other variance-reduction methods, we note that in small-sample settings (e.g., AIME with 30 problems), Bayesian approaches can be highly sensitive to prior specification, and their performance may vary substantially without a well-justified prior. A fair comparison would therefore require careful design and tuning. We will clarify this point in the revision and consider a more systematic comparison in future work.

---

> > ### Author Rebuttal · Reviewer_AT79 · 2026-03-31
> >
> > The rebuttal properly addresses my concerns specifically and clarified the issues that were vague for me well. I maintain my score.

---

> > > ### Author Response · Authors · 2026-04-02
> > >
> > > Dear Reviewer AT79,
> > >
> > > Thank you for your follow-up and for the positive assessment. We are glad the additional details helped clarify the points you raised. We will add more discussion about $\hat\tau$ in the revision thanks to your comment.

---

### Decision · Program_Chairs · 2026-04-30

**Decision:**

Accept (regular)

**Comment:**

This paper addresses the problem of noisy LLM evaluation on math small benchmarks, such as GPQA Diamond, AIME 2025, and GSM8K. The main idea behind this work is to utilize auxiliary comparison signals — having models judge pairs of solution chains via a Bradley Terry model — and use these as control variates within a semiparametric framework to score benchmarks. Because the evaluator's comparison distribution is known by design (the LLM generates it), the estimator achieves the semiparametric efficiency bound, guaranteeing optimal variance reduction.

The underlying statistical machinery — efficient influence functions (EIF) — is well-established in the semiparametric statistics and causal inference literature, but its application to LLM evaluation is new and useful. Reviewers found the framework clean and the experiments convincing (AT79, jGN3, khdH). While the most theory-oriented reviewer (1sUu) noted the variance reduction idea is "nearly trivial" once the formulation is clear, the practical value of bringing these techniques to the LLM evaluation community — where they are not yet standard — is evidenced by the enthusiastic reception from the other reviewers.

The review process provided some genuine improvements to the paper and made it clear that the authors are approaching this work with a great deal of rigor. There was some confusion around the notation (1sUu) which had been mostly resolved. jGN3 raised several practical concerns; the authors responded with new experiments, and jGN3 raised from 4 to 5.

While framed around mathematical reasoning (where ground truth enables clean validation), the framework is fairly generic. The authors note in their conclusion that the methodology accommodates arbitrary auxiliary signals and extends to other performance metrics, such as pass@k accuracy," they do not demonstrate this experimentally. I look forward to seeing how this work can influence evaluation of LLMs more broadly.

Post-rebuttal scores are 5, 4, 5, 5 (mean 4.75), among the most highly rated papers. I recommend accept.